# Non-negative Tensor Mixture Learning for Discrete Density Estimation

## Abstract

We present an expectation-maximization (EM) based unified framework for non-negative tensor decomposition that optimizes the Kullback-Leibler divergence. To avoid iterations in each M-step and learning rate tuning, we establish a general relationship between low-rank decomposition and many-body approximation. Using this connection, we exploit that the closed-form solution of the many-body approximation updates all parameters simultaneously in the M-step. Our framework offers not only a unified methodology for a variety of low-rank structures, including CP, Tucker, and Train decompositions, but also their combinations, forming mixtures of low-rank tensors. The weights of each low-rank tensor in the mixture can be learned from the data, which eliminates the need to carefully choose a single low-rank structure in advance. We empirically demonstrate that our framework provides superior generalization for discrete density estimation compared to conventional tensor-based approaches.

## 1 Introduction

Tensors are versatile data structures used in broad fields such as signal processing (Sidiropoulos et al., 2017), computer vision (Panagakis et al., 2021), and data mining (Papalexakis et al., 2016). It is an established fact that features can be extracted from tensor-formatted data by low-rank decomposition, which approximates the tensor by a linear combination of a few bases (Cichocki et al., 2016; Liu et al., 2022). There are numerous variations of tensor low-rank decompositions, such as CP (Hitchcock, 1927), Tucker (Tucker, 1966), and Tensor Train decompositions (Oseledets, 2011), which differ in the low-rank structure of the decomposed representation.

A series of recent studies (Glasser et al., 2019; Novikov et al., 2021) show that tensor low-rank decomposition is also useful for discrete density estimation, which is an interesting application that takes advantage of the discreteness of the tensor indices. Specifically, given observed discrete samples $x^{(1)}, \dots, x^{(N)}$, the normalized histogram or empirical distribution $p(x)$ can be regarded as a non-negative normalized tensor $\mathcal{T}$, called an *empirical tensor*, and its low-rank reconstruction $\mathcal{P}$ approximates the true distribution as seen in Figure 1. The obtained density can then be used for multiple purposes such as predicting new data points, inferring missing values, or performing outlier detection (Scott, 2015); however, two challenges remain in these current works.

The first challenge is to develop a unified formulation of nonnegative tensor decomposition that works with various kinds of low-rank structures, optimizing the Kullback–Leibler (KL) divergence, which is a natural measure of similarity between probability distributions. In contrast to the well-established SVD-based methods for real- and complex-valued tensor networks (Iblisdir et al., 2007; Román, 2014; Cheng et al., 2019), a general framework for nonnegative tensor decompositions optimizing the KL divergence is not well developed. Consequently, some existing studies of tensor-based density estimation have been performed via optimization of the Frobenius norm (Kargas et al., 2018; Dolgov et al., 2020; Novikov et al., 2021). In addition, due to the lack of a unified KL-divergence-based framework, users must either perform the decompositions with only low-rank structures that have already been developed in a piecemeal manner (Kim et al., 2008; Chi & Kolda, 2012) or differentiate the cost function for the target low-rank structure by themselves. A principled approach that allows users to try various low-rank structures more freely is, therefore, desirable.

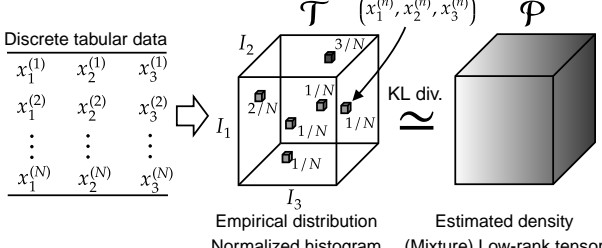

Figure 1: A discrete density estimation by $N$ samples $\boldsymbol{x}^{(1)}, \ldots, \boldsymbol{x}^{(N)}$ for $\boldsymbol{x}^{(n)} = (x_1^{(n)}, x_2^{(n)}, x_3^{(n)})$ and $x_d^{(n)} \in [I_d]$. The normalized histogram, or empirical distribution $p(\boldsymbol{x})$, is identical to a non-negative normalized tensor $\mathcal{T}$, and the true distribution is estimated by its low-rank approximation $\mathcal{P}$.

The second challenge is scalability, as the size of the tensor — which corresponds to the sample space size of the discrete distribution — increases exponentially with the number of features $D$, namely a tensor has $I^D$ elements for the degree of freedom of each feature $I$. However, the number of samples available for training $N$ — typically corresponds to the number of nonzero values of the empirical tensor — is often limited, also deemed the curse of dimensionality. Therefore, it is desirable to develop scalable tensor decomposition that works with high-dimensional data despite a limited number of samples.

To address these two challenges, this paper proposes a unified non-negative tensor factorization method based on the expectation-maximization (EM) algorithm (Dempster et al., 1977). The EM algorithm is an iterative framework for maximum likelihood estimation that repeatedly maximizes the lower bound of the log-likelihood function in two steps, the E-step and the M-step. The naive EM-based formulation for general non-negative tensor decomposition involves either alternating optimization to bound the log-likelihood (E-step) and maximizing each factor individually (M-step), or using gradient-based methods in each M-step. The former approach requires tailoring methods for each type of low-rank structure, while the latter is computationally expensive because of the additional iteration for the gradient method inside the EM iteration. To overcome this issue, we exploit that the optimization in each M-step coincides with a tensor many-body approximation (Ghalamkari et al., 2023) that decomposes tensors by a representation with reduced interactions among tensor modes. We derive the exact closed-form solution for many-body approximation that appears in the M-step for Tucker and Train decomposition, and thereby successfully remove the gradient method in the M-step for various kinds of low-rank decomposition by combining these two formulas.

Our framework inherits the properties of the EM algorithm (Jeff Wu, 1983), and therefore, it always converges regardless of the choice of the low-rank structure assumed in the model. Furthermore, the tensor to be approximated in the M-step is sparse as it is defined in terms of elementwise multiplications by the empirical tensor. As a result, the computational complexity of the proposed method is proportional to the number of samples $N$. Notably, as the EM algorithm is frequently used for maximum likelihood estimation of mixture models, the proposed method allows for density estimation with mixtures of low-rank tensors providing flexible modeling. The mixture model automatically finds appropriate weights for mixed low-rank structures, eliminating the need for the user to define a single low-rank structure in advance. Moreover, the flexibility allows us to mix a low-rank tensor with a constant tensor, which incorporates the noise of data and stabilizes learning. We empirically show that mixture low-rank modeling provides better generalization than pure low-rank tensor models for discrete density estimation. We summarize our contribution as follows:

- We reveal a relationship between tensor many-body approximation and low-rank decomposition.
- Using this relationship, we provide a unified EM-based framework for non-negative low-rank decomposition optimizing the KL-divergence, notably providing simultaneous closed-form updates for all parameters in the M-step while exploiting the sparsity of the observed histogram.
- Based on the proposed framework, we develop a mixture of low-rank tensor modeling that empirically demonstrates inferential robustness and improved generalization.

In the remainder of this section, we describe the problem settings, followed by the definition of many-body approximation, which forms the foundation of this study.

**Problem setup** We construct a normalized histogram, or empirical distribution, from given tabular data with $D$ categorical features. This histogram is identical to a normalized $D$-th order tensor $\mathcal{T} \in \mathbb{R}_{\geq 0}^{I_1 \times \cdots \times I_D}$ where $I_d$ is the degree of freedom (i.e., categories) of the $d$-th categorical feature. To estimate the discrete probability distribution underlying the data, we approximate the tensor $\mathcal{T}$

with a low-rank tensor considering the CP, Tucker, and Tensor Train formats, or their mixture as a convex linear combination of low-rank tensors. The setup for $D = 3$ is shown in Figure 1. The definition of low-rank formats is introduced in Section 3.2.

**Many-body approximation for tensors** The many-body approximation decomposes the tensor by the interactions among the modes described by the interaction diagram. We show an example of the diagram for a fourth-order tensor in Figure 2 where each node (circle) corresponds to a tensor mode and each edge through a black square, ■, denotes the existence of interaction. For a given tensor $\mathcal{T}$, the approximation corresponding to the diagram can be written as

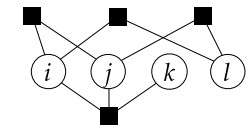

Figure 2: An example of interaction representation

$\mathcal{T}_{ijkl} \simeq \mathcal{P}_{ijkl} = \mathcal{A}_{ijk} B_{ij} C_{il} D_{jl}$ where matrices $B, C$ and $D$ define two-body interactions and the tensor $\mathcal{A}$ defines a three-body interaction. The many-body approximation always provides a globally optimal solution $\mathcal{P}$ that minimizes the KL divergence from the tensor $\mathcal{T}$.

## 2 RELATED WORKS

The EM algorithm is widely used to train models with hidden variables (Dempster et al., 1977). We apply the EM algorithm to tensor decomposition by regarding tensor indices as visible variables and ranks as hidden variables. A lot of studies have shown that decomposing an empirical tensor constructed from observed samples can be used to estimate the underlying distribution behind the data (Kargas et al., 2018; Glasser et al., 2019; Ibrahim & Fu, 2021; Vora et al., 2021). For density estimation, the KL divergence is a natural choice for the objective function of tensor decomposition, and the multiplicative updates methods (MU) are often used to find the low-rank tensor optimizing the KL divergence from an empirical tensor (Kim et al., 2008; Phan & Cichocki, 2008), while the EM-based method has not been established except for the CP decomposition (Huang & Sidiropoulos, 2017; Yeredor & Haardt, 2019; Chege et al., 2022). Our work provides EM-based decomposition for various low-rank structures and their mixtures and enables the update of all parameters simultaneously, which differs from the MU methods that alternatingly update a specific set of parameters while keeping the other parameters fixed. Although a mixture model with the same low-rank structures is considered in (Wu et al., 2023), our approach incorporates a mixture of different low-rank structures, which is a more general framework. Recently, density estimation methods using second and third-order marginals have been developed (Kargas & Sidiropoulos, 2017; Ibrahim & Fu, 2021; Grelier et al., 2022). However, these methods are so far limited to considering the CP decomposition-based model and typically require hyper-parameters for the gradient method, whereas the convergence of the algorithms has not been fully discussed. Contrarily, our approach directly applies to various low-rank structures with hyper-parameter-free optimization and guarantees monotonically decreasing error functions and convergence.

In probabilistic tensor decomposition, tensor elements are sampled from a distribution $p_\theta$ and the model parameter $\theta$ is optimized via the EM algorithm (Kohei et al., 2010; Yılmaz & Cemgil, 2010; Rai et al., 2015). We note that our setting is different from theirs because we do not assume any distribution behind each element of the tensor. Further, the EM algorithm is often used for tensor completion, treating missing values as hidden variables (Tomasi & Bro, 2005; Liu et al., 2015; Song et al., 2019). This task assumes $\mathcal{T}_i = \mathcal{P}_i$ where $\mathcal{T}$ is the given tensor including missing values, $\mathcal{P}$ is the reconstructed low-rank tensor, and $i$ is an index on observed elements of $\mathcal{T}$. Density estimation does not impose this constraint. It has also been reported that the EM algorithm can be applied to sum-product networks (Desana & Schnörr, 2016) by regarding sum-nodes as hidden variables (Peharz et al., 2016). Interestingly, some tensor networks can be represented as sum-product networks (Loconte et al., 2023; 2024) and we therefore expect our approach to generalize to the area of sum-product networks.

## 3 LOW-RANK APPROXIMATION AND MANY-BODY APPROXIMATION

The naive EM-based formulation for non-negative low-rank approximation, which bounds the likelihood in the E-step and optimizes parameters in the M-step, typically relies on an iterative gradient method during the M-step. However, interestingly, we point out that the M-step can be

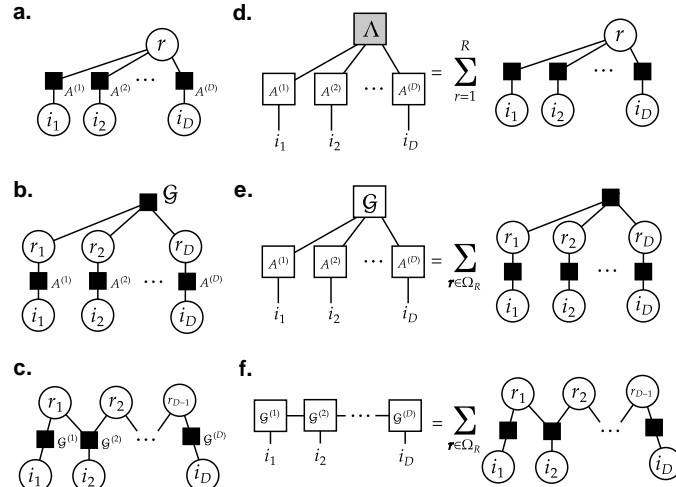

Figure 3: (**a**,**b**,**c**) Interaction diagrams for tensors $\mathcal{Q}^{\mathrm{CP}}, \mathcal{Q}^{\mathrm{Tucker}}$, and $\mathcal{Q}^{\mathrm{Train}}$, respectively, in Equation (1). Each node represents a tensor mode, and the black square, ■, represents the interaction between modes. (**d**,**e**,**f**) Tensor networks for $\mathcal{P}^{\mathrm{CP}}$, $\mathcal{P}^{\mathrm{Tucker}}$, and $\mathcal{P}^{\mathrm{Train}}$, respectively, in Equation (6). Nodes represent factor tensors, and edges connecting nodes represent mode products. The gray-filled $\Lambda$ in (**d**) is a tensor whose hyper-diagonal elements are 1 otherwise 0. These low-rank tensors correspond to the many-body approximation with hidden variables $\boldsymbol{r}$.

regarded as a many-body approximation for a higher-order tensor, and we can eliminate the gradient method in the M-step by the closed-form solution of the many-body approximation derived below.

In the following, we identify a normalized nonnegative tensor with a discrete distribution. Specifically, the tensor element $\mathcal{T}_{i_1,\ldots,i_D}$ is regarded as the value of the distribution $p(x_1 = i_1, \ldots, x_D = i_D)$.

## 3.1 Many-body approximation with exact closed-form solution

As a preliminary step towards a unified low-rank learning framework in Section 4, we here consider the three kinds of many-body approximation, namely a $(D+1)$-th order tensor $\mathcal{Q}^{\mathrm{CP}} \in \mathbb{R}_{\geq 0}^{I_1 \times \cdots \times I_D \times R}$, a $2D$-th order tensor $\mathcal{Q}^{\mathrm{Tucker}} \in \mathbb{R}_{\geq 0}^{I_1 \times \cdots \times I_D \times R_1 \times \cdots \times R_D}$, and a $(2D-1)$-th order tensor $\mathcal{Q}^{\mathrm{Train}} \in \mathbb{R}_{\geq 0}^{I_1 \times \cdots \times I_D \times R_1 \times \cdots \times R_{D-1}}$. When their interactions are described as Figure 3(**a**), (**b**), and (**c**) respectively, they can be factorized as

$$\mathcal{Q}^{\mathrm{CP}}_{i_1\ldots i_D r} = \prod_{d=1}^{D} A^{(d)}_{i_d r}, \quad \mathcal{Q}^{\mathrm{Tucker}}_{i_1\ldots i_D r_1\ldots r_D} = \mathcal{G}_{r_1\ldots r_D} \prod_{d=1}^{D} A^{(d)}_{i_d r_d}, \quad \mathcal{Q}^{\mathrm{Train}}_{i_1\ldots i_D r_1\ldots r_{D-1}} = \prod_{d=1}^{D} \mathcal{G}^{(d)}_{r_{d-1} i_d r_d}, \tag{1}$$

where $i_d \in [I_d]$ and $r_d \in [R_d]$ for $d = 1, \ldots, D$. For simplicity, we suppose $r_0 = r_D = 1$ for $\mathcal{Q}^{\mathrm{Train}}$. Let tensor indices $i_1, \ldots, i_D$ and $r_1, \ldots, r_V$ be $\boldsymbol{i}$ and $\boldsymbol{r}$, respectively, where the integer $V$ is 1 for $\mathcal{Q}^{\mathrm{CP}}$, $D$ for $\mathcal{Q}^{\mathrm{Tucker}}$, and $D-1$ for $\mathcal{Q}^{\mathrm{Train}}$. We denote the domains of $\boldsymbol{i}$ and $\boldsymbol{r}$ by $\Omega_I$ and $\Omega_R$, respectively, i.e., $\boldsymbol{i} \in \Omega_I = [I_1] \times \cdots \times [I_D]$ and $\boldsymbol{r} \in \Omega_R = [R_1] \times \cdots \times [R_V]$. The symbol $\Omega$ with upper indices refers to the index set for all indices other than the upper indices, e.g.,

$$\Omega_I^{\backslash d} = [I_1] \times \cdots \times [I_{d-1}] \times [I_{d+1}] \times \cdots \times [I_D],$$
$$\Omega_R^{\backslash d,d-1} = [R_1] \times \cdots \times [R_{d-2}] \times [R_{d+1}] \times \cdots \times [R_V].$$

Many-body approximation parameterizes tensors as discrete distributions, where the random variables correspond to the tensor modes, and the sample space corresponds to the index set of a tensor. Maximum likelihood estimation finds the globally optimal tensor that minimizes the KL divergence from a given tensor $\mathcal{M}$ in the model space $\mathcal{B}$, which is the set of tensors with specific interactions. Thus, for a given tensor $\mathcal{M}$, the many-body approximation based on the above three interactions maximizes

$$L_{\mathrm{MBA}}(\mathcal{M}; \mathcal{Q}^k) = \sum_{\boldsymbol{i} \in \Omega_I} \sum_{\boldsymbol{r} \in \Omega_R} \mathcal{M}_{\boldsymbol{ir}} \log \mathcal{Q}^k_{\boldsymbol{ir}}, \quad k \in \{\mathrm{CP,\ Tucker,\ Train}\}. \tag{2}$$

This optimization is guaranteed to be a convex problem regardless of the choice of interaction. While the conventional method finds a numerical solution of general many-body approximation by the

Natural gradient method, it has been shown in (Huang & Sidiropoulos, 2017; Yeredor & Haardt, 2019) that the following factors globally maximize $L_{\text{MBA}}(\mathcal{M}; \mathcal{Q}^{\text{CP}})$:

$$A_{i_d r}^{(d)} = \frac{\sum_{\boldsymbol{i} \in \Omega_I^{\backslash d}} \mathcal{M}_{\boldsymbol{ir}}}{\mu^{1/D} \left( \sum_{\boldsymbol{i} \in \Omega_I^{\backslash d}} \mathcal{M}_{\boldsymbol{ir}} \right)^{1-1/D}}, \quad \mu = \sum_{\boldsymbol{i} \in \Omega_I} \sum_{\boldsymbol{r} \in \Omega_R} \mathcal{M}_{\boldsymbol{ir}}, \tag{3}$$

which is consistent with a mean-field approximation (Ghalamkari & Sugiyama, 2021; 2023). As a generalization of the above result, we presently provide the optimal solution of the many-body approximation of $\mathcal{Q}^{\text{Tucker}}$ and $\mathcal{Q}^{\text{Train}}$ in closed-form. The following tensors globally maximizes $L_{\text{MBA}}(\mathcal{M}; \mathcal{Q}^{\text{Tucker}})$:

$$\mathcal{G}_{\boldsymbol{r}} = \frac{\sum_{\boldsymbol{i} \in \Omega_I} \mathcal{M}_{\boldsymbol{ir}}}{\sum_{\boldsymbol{i} \in \Omega_I} \sum_{\boldsymbol{r} \in \Omega_R} \mathcal{M}_{\boldsymbol{ir}}}, \quad A_{i_d r_d}^{(d)} = \frac{\sum_{\boldsymbol{i} \in \Omega_I^{\backslash d}} \sum_{\boldsymbol{r} \in \Omega_R^{\backslash d}} \mathcal{M}_{\boldsymbol{ir}}}{\sum_{\boldsymbol{i} \in \Omega_I} \sum_{\boldsymbol{r} \in \Omega_R^{\backslash d}} \mathcal{M}_{\boldsymbol{ir}}}, \tag{4}$$

and the following tensors globally maximizes $L_{\text{MBA}}(\mathcal{M}; \mathcal{Q}^{\text{Train}})$:

$$\mathcal{G}_{r_{d-1} i_d r_d}^{(d)} = \frac{\sum_{\boldsymbol{i} \in \Omega_I^{\backslash d}} \sum_{\boldsymbol{r} \in \Omega_R^{\backslash d, d-1}} \mathcal{M}_{\boldsymbol{ir}}}{\sum_{\boldsymbol{i} \in \Omega_I} \sum_{\boldsymbol{r} \in \Omega_R^{\backslash d}} \mathcal{M}_{\boldsymbol{ir}}}, \tag{5}$$

where we assume $r_0 = r_D = 1$. We formally derive the above closed-form solutions in Theorems 2 and 3 in the supplementary material. Notably, we can obtain exact solutions also for more complicated many-body approximations by combining these solutions, which we discuss in Section 4.3.

### 3.2 Low-rank approximation and Many-body approximation

When summing the tensors in Equation (1) over indices $\boldsymbol{r}$, these models are identical to the traditional low-rank models, CP, Tucker, and Tensor Train formats, respectively.

$$\mathcal{P}_{\boldsymbol{i}}^{\text{CP}} = \sum_r \mathcal{Q}_{\boldsymbol{ir}}^{\text{CP}}, \quad \mathcal{P}_{\boldsymbol{i}}^{\text{Tucker}} = \sum_{r_1 \dots r_D} \mathcal{Q}_{\boldsymbol{ir_1 \dots r_D}}^{\text{Tucker}}, \quad \mathcal{P}_{\boldsymbol{i}}^{\text{Train}} = \sum_{r_1 \dots r_{D-1}} \mathcal{Q}_{\boldsymbol{ir_1 \dots r_{D-1}}}^{\text{Train}}. \tag{6}$$

Since many-body approximation treats the tensor indices as discrete random variables and variables $\boldsymbol{r}$ in Equation (6) are marginalized, we consider these low-rank tensors as models in which the random variables $\boldsymbol{i}$ and $\boldsymbol{r}$ represent visible and hidden variables, respectively. The degree of freedom of hidden variables $(R_1, \dots, R_V)$ corresponds to *CP rank*, *Tucker rank*, and *train rank* with $V = 1$ for $\mathcal{Q}^{\text{CP}}$, $V = D$ for $\mathcal{Q}^{\text{Tucker}}$, and $V = D-1$ for $\mathcal{Q}^{\text{Train}}$, respectively. Since any low-rank factorization decomposes a tensor by summing over its ranks, *any low-rank approximation can be regarded as a many-body approximation with hidden variables*. When a low-rank tensor $\mathcal{P}$ is obtained by marginalization of a tensor $\mathcal{Q}$ with appropriately selected modes, we refer to tensor $\mathcal{Q}$ as the *low-body tensor* corresponding to $\mathcal{P}$. For example, $\mathcal{Q}^{\text{CP}}$, $\mathcal{Q}^{\text{Tucker}}$ and $\mathcal{Q}^{\text{Train}}$ are low-body tensors corresponding to a low CP-rank tensor $\mathcal{P}^{\text{CP}}$, low Tucker-rank tensor $\mathcal{P}^{\text{Tucker}}$, and low Train-rank tensor $\mathcal{P}^{\text{Train}}$, respectively. In this paper, hidden variables — corresponding to modes summed according to the low-rank structure — are denoted by $\boldsymbol{r}$ and visible variables — variables other than hidden variables — by $\boldsymbol{i}$.

The many-body approximation involves only visible variables in the model, making maximum likelihood estimation a convex optimization problem, while the low-rank approximation requires a nonconvex optimization due to the hidden variables in the model. Therefore, although finding an exact solution for the low-rank approximation is challenging, the optimization remains tractable through the EM algorithm (Dempster et al., 1977), a well-known general framework for maximum likelihood estimation that accommodates hidden variables.

## 4 Discrete density estimation via non-negative tensor learning

Based on the exact solutions of many-body approximation derived in Section 3, we develop a novel framework *EM non-negative tensor learning* for discrete density estimation. Our framework has two advantages: (1) it achieves linear computational complexity relative to the number of nonzero elements in the input tensor, as described in Section 4.2; and (2) it offers the flexibility to incorporate various low-rank structures, such as CP, Tucker, Train, their mixtures, and adaptive noise terms, while preserving the convexity of E-step and M-step, as detailed in Sections 4.3 and 4.4. Neither of these advantages has been explored in previous studies (Kim et al., 2008; Chi & Kolda, 2012).

## 4.1 EM-METHOD FOR MIXTURE OF LOW-RANK APPROXIMATIONS

Here, we provide a unified EM-based method for low-rank approximations. We maximize the negative cross-entropy from a given $D$-th order tensor $\mathcal{T}$ to a mixture of $K$ low-rank tensors $\mathcal{P}^1, \ldots, \mathcal{P}^K$,

$$L(\hat{\mathcal{Q}}) = \sum_{\boldsymbol{i} \in \Omega_I} \mathcal{T}_{\boldsymbol{i}} \log \sum_{k=1}^K \eta^k \mathcal{P}_{\boldsymbol{i}}^k, \qquad \mathcal{P}_{\boldsymbol{i}}^k = \sum_{\boldsymbol{r} \in \Omega_{R^k}} \mathcal{Q}_{\boldsymbol{ir}}^k, \tag{7}$$

where the mixture ratio satisfies $\sum_k \eta^k = 1$ and $\eta^k \geq 0$ for all $k \in [K]$ and setting $K = 1$ yields a conventional low-rank tensor decomposition. Each low-rank tensor $\mathcal{P}^k$ is normalized, i.e., $\sum_{\boldsymbol{i}} \mathcal{P}_{\boldsymbol{i}}^k = 1$. We let the model $\mathcal{P}$ be the convex linear combination of low-rank tensors, that is, $\mathcal{P} = \sum_{k=1}^K \eta^k \mathcal{P}^k$. We denote the number of hidden variables in the tensor $\mathcal{Q}^k$ by $V^k$, and $\boldsymbol{r} \in \Omega_{R^k} = [R_1^k] \times \cdots \times [R_{V^k}^k]$. For simplicity, we introduce $\hat{\mathcal{Q}}_{\boldsymbol{ir}}^k = \eta^k \mathcal{Q}_{\boldsymbol{ir}}^k$, and refer to the tensors $(\hat{\mathcal{Q}}^1, \ldots, \hat{\mathcal{Q}}^K)$ as $\hat{\mathcal{Q}}$. We apply Jensen's inequality (Jensen, 1906) to the objective function $L(\hat{\mathcal{Q}})$ in order to move the summation over hidden variables $\boldsymbol{r}$ outside the logarithm function thereby obtaining the lower bound,

$$L(\hat{\mathcal{Q}}) \geq \overline{L}(\hat{\mathcal{Q}}, \Phi) = \sum_{\boldsymbol{i} \in \Omega_I} \sum_{k=1}^K \sum_{\boldsymbol{r} \in \Omega_{R^k}} \mathcal{T}_{\boldsymbol{i}} \Phi_{\boldsymbol{ir}}^k \log \frac{\hat{\mathcal{Q}}_{\boldsymbol{ir}}^k}{\Phi_{\boldsymbol{ir}}^k}, \tag{8}$$

for any $K$ tensors $\Phi = (\Phi^1, \ldots, \Phi^K)$ where $\Phi^k$ is a $(D + V^k)$-th order tensor satisfying $\sum_k \sum_{\boldsymbol{r} \in \Omega_{R^k}} \Phi_{\boldsymbol{ir}}^k = 1$. The derivation of the inequality (8) is described in Proposition 1 in the supplementary material. The above lower bound can be decoupled into independent multiple many-body approximations for tensors $\mathcal{M}^1, \ldots, \mathcal{M}^K$ where each tensor $\mathcal{M}^k$ is defined as $\mathcal{M}_{\boldsymbol{ir}}^k = \mathcal{T}_{\boldsymbol{i}} \Phi_{\boldsymbol{ir}}^k$, and an optimization problem for the mixture ratio $\eta = (\eta^1, \ldots, \eta^K)$. More specifically, we decouple the lower bound as follows:

$$\overline{L}(\hat{\mathcal{Q}}, \Phi) = \sum_{k=1}^K L_{\mathrm{MBA}}(\mathcal{M}^k; \mathcal{Q}^k) + J(\eta), \quad J(\eta) = \sum_{\boldsymbol{i} \in \Omega_I} \sum_{k=1}^K \sum_{\boldsymbol{r} \in \Omega_{R^k}} \mathcal{T}_{\boldsymbol{i}} \Phi_{\boldsymbol{ir}}^k \log \eta^k, \tag{9}$$

where the objective function of many-body approximation $L_{\mathrm{MBA}}$ is introduced in Equation (2). The EM algorithm iteratively optimizes the lower bound for tensors $\Phi$ in the E-step and tensors $\hat{\mathcal{Q}}$ in the M-step until convergence. Each step is a convex optimization while Equation (7) is a non-convex function. This procedure is guaranteed to converge, and each iteration increases the objective function (7) monotonically, which we prove in Theorem 4 in the supplementary material while the general convergence theorem of the MU method is still an open problem, requiring proof of convergence for each minor change in the objective function, such as varying the low-rank structure or adding regularization terms.

**E-step** We maximize the lower bound $\overline{L}(\hat{\mathcal{Q}}, \Phi)$ for tensors $\Phi = (\Phi^1, \ldots, \Phi^K)$, that is,

$$\Phi = \arg\max_{\Phi \in \boldsymbol{\mathcal{D}}} \overline{L}(\hat{\mathcal{Q}}, \Phi),$$

where the solution space $\boldsymbol{\mathcal{D}}$ is a tuple of $K$ tensors such that each tensor is normalized by hidden variables, i.e., $\boldsymbol{\mathcal{D}} = \left\{ \left( \Phi^1, \ldots, \Phi^K \right) \mid \sum_k \sum_{\boldsymbol{r} \in \Omega_{R^k}} \Phi_{\boldsymbol{ir}}^k = 1 \right\}$. The optimal solution $\Phi^k$ is

$$\Phi_{\boldsymbol{ir}}^k = \frac{\hat{\mathcal{Q}}_{\boldsymbol{ir}}^k}{\sum_{k=1}^K \sum_{\boldsymbol{r} \in \Omega_{R^k}} \hat{\mathcal{Q}}_{\boldsymbol{ir}}^k}, \tag{10}$$

as shown in Proposition 2 in the supplementary material. The denominator in Equation (10) is equivalent to the definition of the model $\mathcal{P}$.

**M-step** We maximize the lower bound $\overline{L}(\hat{\mathcal{Q}}, \Phi)$ for tensors $\mathcal{Q} = (\mathcal{Q}^1, \ldots, \mathcal{Q}^K)$ and non-negative weights $\eta$. Since the lower bound can be decoupled as shown in Equation (9), the required optimizations in the M-step are as follows:

$$\mathcal{Q}^k = \arg\max_{\mathcal{Q}^k \in \boldsymbol{\mathcal{B}}^k} L_{\mathrm{MBA}}(\mathcal{M}^k; \mathcal{Q}^k), \quad \eta = \arg\max_{0 \leq \eta^k \leq 1, \sum_k \eta^k = 1} J(\eta), \tag{11}$$

---

**Algorithm 1:** Non-negative Tensor Mixture Learning

---

**input** : Non-negative tensor $\mathcal{T}$, the number of mixtures $K$, and ranks $(R^1, \dots, R^K)$
Initialize $\mathcal{Q}^k$ and $\eta^k$ for all $k \in [K]$;
**repeat**
    $\mathcal{P}_{\boldsymbol{i}} \leftarrow \sum_k \eta^k \mathcal{P}_{\boldsymbol{i}}^k$ where $\mathcal{P}_{\boldsymbol{i}}^k = \sum_{\boldsymbol{r} \in R^k} \mathcal{Q}_{\boldsymbol{ir}}^k$;
    $\mathcal{M}_{\boldsymbol{ir}}^k \leftarrow \mathcal{T}_{\boldsymbol{i}} \mathcal{Q}_{\boldsymbol{ir}}^k / \mathcal{P}_{\boldsymbol{i}}$ for all $k \in [K]$;        // E-step
    Update tensor $\mathcal{Q}^k$ for all $k \in [K]$ using Equations (1), (4), and (5);    // M-step
    Update mixture ratio $\eta^k$ using Equation (12) for all $k \in [K]$;    // M-step
**until** *Convergence*;
**return** $\mathcal{P}$

---

where the model space $\boldsymbol{\mathcal{B}}^k$ is the set of low-body tensors corresponding to the $k$-th low-rank tensor $\mathcal{P}^k$. The naive implementation of the M-step requires a gradient method to solve the many-body approximation for $\mathcal{M}^k$. The existing algorithm for many-body approximation is based on the Natural gradient method (Amari, 2016), which requires cubic computational complexity for the number of parameters in the low-body tensor $\mathcal{Q}^k$. Thus, repeating the gradient method in each M-step is computationally expensive. However, we introduced the closed-form solution of the many-body approximation in Section 3.1, which eliminates the iterative gradient method in the M-step for Tucker and Train decomposition. We discuss more complicated low-rank structures in Section 4.3.

We also provide the optimal update rule for the mixture ratio $\eta$ in closed form. By the condition $\partial J(\eta)/\partial \eta^k = 0$ and the normalizing condition $\sum_k \eta^k = 1$, the optimal $\eta^k$ is simply given as follows:

$$\eta^k = \frac{\sum_{\boldsymbol{i} \in \Omega_I} \sum_{\boldsymbol{r} \in \Omega_{R^k}} \mathcal{M}_{\boldsymbol{ir}}^k}{\sum_{k=1}^{K} \sum_{\boldsymbol{i} \in \Omega_I} \sum_{\boldsymbol{r} \in \Omega_{R^k}} \mathcal{M}_{\boldsymbol{ir}}^k} \tag{12}$$

which is shown in Proposition 3 in the supplementary material. It is straightforward to check that the M-step is a convex optimization problem whatever the low-rank structure assumed on $\mathcal{P}^k$ since we can decouple the lower bound into multiple independent convex many-body approximations.

While the EM-based method for CP decomposition optimizing the KL divergence has already been developed in (Huang & Sidiropoulos, 2017; Yeredor & Haardt, 2019), our proposed approach addresses more general low-rank decompositions and their mixtures. We provide the entire algorithm in Algorithm 1. Notably, the framework does not require a learning rate that needs to be carefully tuned when relying on gradient-based methods. Moreover, the proposed method updates all parameters simultaneously in closed form, which differs from the multiplicative update rule such as (Kim et al., 2008). The extension to non-normalized non-negative tensors is described in Section B.2 in the supplemental material.

## 4.2 Analysis of computational complexity

In the following, we discuss the computational complexity of the proposed EM low-rank approximation without a mixture, specifically when $K = 1$. It is straightforward to see that the computational complexity for $K > 1$ is simply the sum of the complexity of each of the EM low-rank approximations used in the mixture.

In our approach, we compute the sum over the visible variables $\boldsymbol{i} \in \Omega_I$ of the tensor $\mathcal{M}^k$ in the M-step, using Equations (4), (5), and (12) for which the computational cost increases rapidly as the size of the tensor increases. However, if the input tensor $\mathcal{T}$ is sparse, which is a reasonable assumption for most density estimation tasks, we can reduce the computational complexity of the summation because the tensor $\mathcal{M}^k$ is also sparse for indices $\boldsymbol{i}$ as the definition of $\mathcal{M}_{\boldsymbol{ir}}^k = \mathcal{T}_{\boldsymbol{i}} \Phi_{\boldsymbol{ir}}^k$. More specifically, we replace the sum over the visible variables $\sum_{\boldsymbol{i} \in \Omega_I}$ with $\sum_{\boldsymbol{i} \in \Omega_I^\circ}$ where $\Omega_I^\circ$ is the set of indices of nonzero values of the tensor $\mathcal{T}$. The resulting time computational complexity is $O(\gamma N D R)$ for EM-CP decomposition and $O(\gamma D N R^D)$ for EM-Tucker and a naive EM-Train decomposition where $N$ is the number of nonzero values, $D$ is the number of modes in the tensor, $\gamma$ is the number of iteration, assuming ranks are $(R, \dots, R)$ for all low-rank models.

---

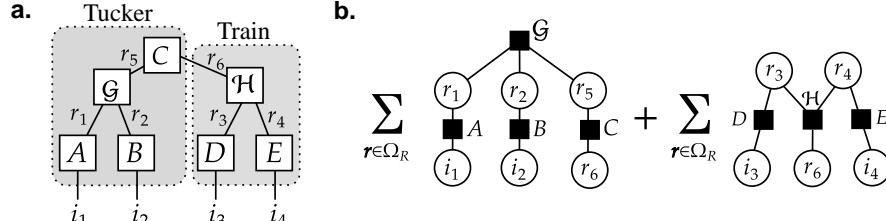

Figure 4: (**a**) A tensor tree structure (**b**) The M-step of the tree structure is decoupled into two solvable many-body approximations, which are enclosed by dotted lines in (**a**).

Furthermore, we can reduce the computational cost of the tensor train decomposition to $O(\gamma DNR^2)$ by computing the sum over the latent variables $\boldsymbol{r} \in \Omega_R$ as follows. Firstly, we introduce the following tensors

$$\mathcal{G}^{(\to d)}_{i_1,\dots,i_d,r_d} = \sum_{r_{d-1}} \mathcal{G}^{(\to d-1)}_{i_1,\dots,i_{d-1},r_{d-1}} \mathcal{G}^{(d)}_{r_{d-1}i_d r_d}, \quad \mathcal{G}^{(d\leftarrow)}_{i_{d+1},\dots,i_D,r_d} = \sum_{r_{d+1}} \mathcal{G}^{(d+1)}_{r_d i_{d+1} r_{d+1}} \mathcal{G}^{(d+1\leftarrow)}_{i_{d+2},\dots,i_D,r_{d+1}}$$

(13)

with $\mathcal{G}^{(\to 1)} = \mathcal{G}^{(1)}$, $\mathcal{G}^{(D-1\leftarrow)} = \mathcal{G}^{(D)}$, and $\mathcal{G}^{(\to 0)} = \mathcal{G}^{(D\leftarrow)} = 1$. Each complexity is $O(R_d)$ to get $\mathcal{G}^{(\to d)}$ and $\mathcal{G}^{(d\leftarrow)}$ when we compute them in the order of $\mathcal{G}^{(\to 2)}, \mathcal{G}^{(\to 3)}, \dots, \mathcal{G}^{(\to D)}$, and $\mathcal{G}^{(D-2\leftarrow)}, \mathcal{G}^{(D-3\leftarrow)}, \dots, \mathcal{G}^{(1\leftarrow)}$, respectively. The low-rank tensor $\mathcal{P}$ can be written as $\mathcal{P} = \mathcal{G}^{(\to D)} = \mathcal{G}^{(0\leftarrow)}$. Then, the update rule can be written as

$$\mathcal{G}^{(d)}_{r_{d-1}i_d r_d} = \frac{\sum_{\boldsymbol{i}\in\Omega^{o\backslash d}_{I,i_d}} \frac{\mathcal{T}_{\boldsymbol{i}}}{\mathcal{P}_{\boldsymbol{i}}} \mathcal{G}^{(\to d-1)}_{i_1,\dots,i_{d-1},r_{d-1}} \mathcal{G}^{(d)}_{r_{d-1}i_d r_d} \mathcal{G}^{(d\leftarrow)}_{i_{d+1},\dots,i_D,r_d}}{\sum_{\boldsymbol{i}\in\Omega^o_I} \frac{\mathcal{T}_{\boldsymbol{i}}}{\mathcal{P}_{\boldsymbol{i}}} \mathcal{G}^{(\to d)}_{i_1,\dots,i_d,r_d} \mathcal{G}^{(d\leftarrow)}_{i_{d+1},\dots,i_D,r_d}}$$

(14)

for $\Omega^{o\backslash d}_{I,i_d} = \Omega^o_I \cap [I_1] \times \dots \times [I_{d-1}] \times \{i_d\} \times [I_{d+1}] \dots \times [I_D]$. We used the relation $\mathcal{M}_{\boldsymbol{i}\boldsymbol{r}} = \mathcal{T}_{\boldsymbol{i}}\Phi_{\boldsymbol{i}\boldsymbol{r}}$ and $\Phi_{\boldsymbol{i}\boldsymbol{r}} = \mathcal{Q}_{\boldsymbol{i}\boldsymbol{r}}/\mathcal{P}_{\boldsymbol{i}}$ to get the above update rule. Not all elements in Equation (13) are necessary for updating tensors by Equation (14). Since the number of elements in $\Omega^o$ is $N$, the resulting complexity is $O(\gamma DNR^2)$. We provide the EM-Train factorization in Algorithm 2 in the supplementary material.

**Reordering tensor modes for EM-Train** The tensor train decomposition results are influenced by the order of the modes in the tensor. To quantify the influence on and potentially enhance the performance of these tensor decompositions, we reorder the tensor modes based on normalized mutual information (NMI) between pairwise features in the data. We describe the definition of NMI in Section B.3 in the supplementary material. To illustrate this, we consider a dataset with five features. First, we select the two modes, $j_1$ and $j_2$, with the highest NMI. These become the middle modes in the rearranged order. Next, we choose the mode $j_3$ with the second-highest NMI with $j_1$, placing it to the left of $j_1$. Then, we select the mode $j_4$ with the highest NMI with $j_2$ among the remaining unselected features (i.e., modes), placing it to the right of $j_2$. This process is repeated until all features (modes) are selected, resulting in the tensor modes being rearranged from $(1, 2, 3, 4, 5)$ to $(j_5, j_3, j_1, j_2, j_4)$. The effectiveness of the reordering is examined in the supplementary material.

### 4.3 EM ALGORITHM FOR MORE GENERAL LOW-RANK STRUCTURES

We now discuss how to find the solution for the many-body approximation required in the M-step when a more complex low-rank structure is assumed in the model. By the basic property of the logarithm function, the function $L_{\mathrm{MBA}}$ to be optimized in the M-step can be decoupled into independent solvable parts with closed form. As an example, we here see the tensor network state described in Figure 4(**a**), which is known as a typical tensor tree structure (Liu et al., 2018). The objective function of many-body approximation in the M-step can be decoupled as follows:

$$L_{\mathrm{MBA}}(\mathcal{M};\mathcal{Q}) = \sum_{\boldsymbol{i}\in\Omega_I} \sum_{\boldsymbol{r}\in\Omega_R} \mathcal{M}_{\boldsymbol{i}\boldsymbol{r}} \log \mathcal{G}_{r_1 r_2 r_5} A_{i_1 r_1} B_{i_2 r_2} C_{r_5 r_6} D_{i_3 r_3} \mathcal{H}_{r_3 r_6 r_4} E_{r_4 i_4}$$

$$= L_{\mathrm{MBA}}(\mathcal{M}^{\mathrm{Tucker}};\mathcal{Q}^{\mathrm{Tucker}}) + L_{\mathrm{MBA}}(\mathcal{M}^{\mathrm{Train}};\mathcal{Q}^{\mathrm{Train}})$$

(15)

Table 1: Negative log-likelihood per sample on the test dataset.

|  | CNMFOPT | MPS | BM | LPS | EMCPTrainON |
|---|---|---|---|---|---|
| SolarFlare | 6.96(0.16) | 6.08(0.05) | 6.00(0.05) | **5.88**(0.01) | 5.98(0.04) |
| SPECT | 11.77(0.03) | 12.57(0.86) | 11.96(0.31) | 11.88(0.45) | **11.60**(0.27) |
| Lympho. | 12.45(0.07) | 13.03(0.04) | 12.34(0.20) | 12.37(0.17) | **12.13**(0.35) |
| Votes | 11.44(0.20) | 13.35(1.78) | 10.45(0.14) | 10.47(0.04) | **10.37**(0.04) |
| Tumor | 9.42(0.09) | 9.19(0.19) | 9.17(0.17) | 9.17(0.04) | **9.11**(0.04) |
| Chess | 14.83(0.16) | 12.45(0.17) | 12.45(0.30) | 11.89(0.05) | **11.22**(0.18) |
| Led7 | 5.62(0.05) | 5.86(0.017) | 5.73(0.31) | 5.16(0.04) | **4.77**(0.01) |
| DMFT | 7.30(0.02) | 7.25(0.05) | 7.26(0.09) | 7.13(0.02) | **7.12**(0.02) |

where we define $\mathcal{M}^{\text{Tucker}}_{i_1 i_2 r_1 r_2 r_5} = \sum_{i_3 i_4 r_3 r_4 r_6} \mathcal{M}_{\boldsymbol{ir}}$ and $\mathcal{M}^{\text{Train}}_{i_3 i_4 r_3 r_4 r_6} = \sum_{i_1 i_2 r_1 r_2 r_5} \mathcal{M}_{\boldsymbol{ir}}$. We can optimize both terms in the final line by the closed-form solution introduced in Section 3.1. We provide theoretical support for this procedure, including normalization conditions, in Section B.1 in the supplementary material. By decoupling the problem with CP, Tucker, and Train decompositions, our approach approximates tensors with a wide variety of low-rank structures.

### 4.4 ADAPTIVE NOISE LEARNING

Our approach deals with not only typical low-rank structures and their mixtures but also regularization and stabilization terms. For example, we here define model $\mathcal{P}$ as the mixture of a low-rank tensor $\mathcal{P}^{\text{low-rank}}$ and a normalized uniform tensor $\mathcal{P}^{\text{noise}}$ whose elements all are $1/|\Omega_I|$ as

$$\mathcal{P} = (1 - \eta^{\text{noise}})\mathcal{P}^{\text{low-rank}} + \eta^{\text{noise}}\mathcal{P}^{\text{noise}},$$

where $|\Omega_I|$ is the number of elements of the tensor $\mathcal{T}$, that is, $|\Omega_I| = I_1 I_2 \ldots I_D$. The value $\eta^{\text{noise}}$ indicates the magnitude of global shift or background noise in the data. This is a learnable parameter from the data, not a hyperparameter. While adding a small uniform constant to a tensor or factors is often used as a heuristic to stabilize learning (Cichocki & Phan, 2009; Gillis & Glineur, 2012), our method provides a principled approach to learning the constant from the data. It is obvious from the discussion in Section 4.1 that the convergence remains even in the presence of the noise term. We show in Section C in the supplementary material that the noise term stabilizes the learning.

## 5 NUMERICAL EXPERIMENTS

We empirically examined the effectiveness of our framework using eight real-world categorical datasets. We downloaded these datasets from the repositories described in Table 7 and divided samples into 70% training, 15% validation, and 15% test samples to form train, validation, and test tensors, respectively. We tuned tensor ranks to minimize the negative log-likelihood (NLL) per sample on the validation tensor and evaluated each model on the test tensor. The experimental setup is detailed in Section D the supplementary material.

We compared our approach to the following baseline methods: Pairwise marginalized method (CNMFOPT),

Table 2: NLL on the test dataset

|  | EMCPTrain | EMCPTrainN |
|---|---|---|
| SolarFlare | 6.34(0.27) | **5.98**(0.06) |
| SPECT | **11.22**(0.11) | 11.49(0.23) |
| Lympho. | 22.13(11.02) | **11.83**(0.41) |
| Votes | 11.13(0.65) | **10.39**(0.12) |
| Tumor | 9.28(0.14) | **9.24**(0.20) |
| Chess | NaN(NaN) | **11.25**(0.17) |
| Led7 | 4.86(0.15) | **4.79**(0.01) |
| DMFT | NaN(NaN) | **7.13**(0.02) |

Matrix Product States (MPS), Born Machine (BM), and Locally Purified State (LPS), which are also tensor-based methods (Ibrahim & Fu, 2021; Glasser et al., 2019). The learning rate of the baseline methods was tuned so that the reconstructed tensor minimizes the KL divergence from the validation data. In contrast, the proposed methods do not require a learning rate. While the proposed framework can explore a variety of low-rank structures and their mixtures, we examine here the performance of EMCPTrainON, a mixture of CP and Train decompositions with adaptive noise term and tensor mode re-ordering. We ran each of the procedures five times with random initialization and reported mean values and the standard error in Table 1. We see that EMCPTrainON has the best generalization performance on all datasets except SolarFlare. Additional comparisons with ten traditional tensor decompositions are also provided in Section C in the supplementary material.

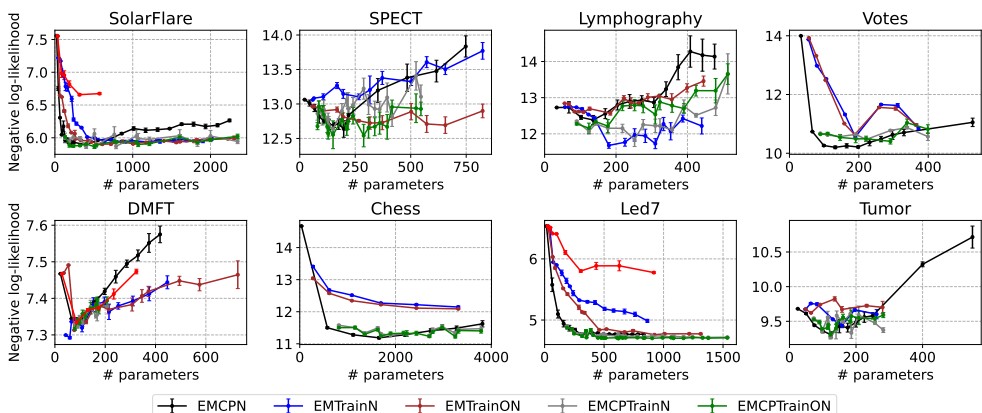

Figure 5: Validation error for each number of parameters for each method. The symbol 'O' indicates that the model includes mode reordering, and 'N' indicates that the model includes the noise term.

Furthermore, we evaluated the validation error of the proposed EMCP, EMTrain, and EMTucker decompositions with the adaptive noise term by varying the number of parameters in Figure 5. For the Tucker decomposition, we only considered three relatively small datasets, SolarFlare, DMFT, and, Led7, due to its high computational cost. No method alone shows superior performance on all datasets, however, we observe that the mixture of CP and Train generally performs well by combining the modeling capabilities of the two decomposition approaches. As a result, for SolarFlare, SPECT, and Led7 datasets, EMCPTrainON achieved the lowest validation error, while EMTrainN had the lowest validation error for Lymphography datasets, and EMCPN had the lowest error for Votes. Notably, our approach readily incorporates all these procedures allowing various low-rank decompositions and their mixture.

We also compared the results using EMCPTrain with and without the adaptive noise term to verify its usefulness in Table 2. Although Theorem 4 guarantees the convergence of the EM algorithm in theory, the objective function often becomes `NaN` when the model has no adaptive noise term. This is because of numerical instability due to extremely small values in the logarithm function. The noise term, which adds values to all elements of the tensor, eliminates this problem. We also provide Figure 8 in the supplementary material showing how the validation errors for models without the adaptive noise term become significantly larger when the number of parameters is large.

The EM algorithm often requires a large number of iterations to converge (Ng et al., 2012). Thus, we conducted additional experiments in Section C.6 and confirmed that the proposed method converges with fewer iterations than the batch gradient method and with a similar number of iterations as the MU methods.

## 6 CONCLUSION

While it is a well-established principle that both real- and complex-valued tensors can be approximated with various low-rank structures based on local singular value decomposition (SVD), a unified framework for non-negative low-rank approximation has not been developed. Consequently, nonnegative tensor factorizations typically require piecemeal tailored implementations or gradient methods. This study introduces an EM-based unified framework that decouples the non-negative low-rank structure into multiple solvable many-body approximations and applies a closed-form solution locally, thereby eliminating the need for gradient methods. Our framework not only uniformly handles various low-rank structures but also supports adaptive noise terms and mixtures of low-rank tensors without losing the convergence guarantee. This flexibility of our approach raises the question of how to systematically determine each component of the mixture, presenting an intriguing direction for future research. Empirical results demonstrate that our framework achieves superior generalization performance compared to baseline methods.

## ETHICS STATEMENT

Our work aims to establish a fundamental methodology for machine learning, and there is no direct risk of misuse or ethical issues.

## REPRODUCIBILITY STATEMENT

The source code for reproducing all experiments is included in the supplementary material, along with a document that explains how to run the code. Proofs of the Theorems and Propositions can be found in Section A. The discussion of the convergence guarantee can be found in Section A.2. Details of the dataset used in the experiments, including the link to download the datasets, are provided in Section D.3. The experimental setup is described in Section D.1, with hyperparameter tuning covered in Section D.2. The computing environment is explained in Section D.3.

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

# Supplementary Material

## Table of Contents

## A  PROOFS

### A.1  PROOFS FOR EXACT SOLUTIONS OF MANY-BODY APPROXIMATION

First, we show the known solution formulas for the best CP rank-1 approximation that globally minimizes the KL divergence from the given tensor.

**Theorem 1** (Optimal M-step in CP decomposition (Huang & Sidiropoulos, 2017))**.** *For a given non-negative tensor $\mathcal{M} \in \mathbb{R}_{\geq 0}^{I_1 \times \cdots \times I_D \times R}$, its many body approximation with interactions as described in Figure 3(**a**) is given as*

$$A_{i_d r}^{(d)} = \frac{\sum_{\boldsymbol{i} \in \Omega_I^{\backslash d}} \mathcal{M}_{\boldsymbol{i}r}}{\mu^{1/D} \left( \sum_{\boldsymbol{i} \in \Omega_I^{\backslash d}} \mathcal{M}_{\boldsymbol{i}r} \right)^{1-1/D}}, \quad \mu = \sum_{\boldsymbol{i} \in \Omega_I} \sum_{r \in \Omega_R} \mathcal{M}_{\boldsymbol{i}r}$$

*Proof.* Please refer to the original paper by Huang & Sidiropoulos (2017).   □

In the following, we provide proofs of the closed-form solutions of many-body approximation in Figure 3(**b**) and (**c**). When a factor in a low-body tensor is multiplied by $\nu$, the value of the objective function of many-body approximation remains the same if another factor is multiplied by $1/\nu$, which we call the *scaling redundancy*. The key idea in the following proofs is reducing the scaling redundancy and absorbing the normalizing conditions of the entire tensor into a single factor. This enables the decoupling of the normalized condition of the entire tensor into independent conditions about factors. This trick is also used in Section B.1 to decouple more complicated low-rank structures into a combination of CP, Tucker, and Train decompositions.

**Theorem 2** (The closed form of the optimal M-step in Tucker decomposition)**.** *For a given tensor* $\mathcal{M} \in \mathbb{R}_{\geq 0}^{I_1 \times \ldots I_D \times R_1 \times \cdots \times R_D}$, *its many-body approximation with interaction described in Figure 3($\boldsymbol{b}$) is given as*

$$\mathcal{G}_{\boldsymbol{r}} = \frac{\sum_{\boldsymbol{i} \in \Omega_I} \mathcal{M}_{\boldsymbol{ir}}}{\sum_{\boldsymbol{i} \in \Omega_I} \sum_{\boldsymbol{r} \in \Omega_R} \mathcal{M}_{\boldsymbol{ir}}}, \quad A_{i_d r_d}^{(d)} = \frac{\sum_{\boldsymbol{i} \in \Omega_I^{\backslash d}} \sum_{\boldsymbol{r} \in \Omega_r^{\backslash d}} \mathcal{M}_{\boldsymbol{ir}}}{\sum_{\boldsymbol{i} \in \Omega_I} \sum_{\boldsymbol{r} \in \Omega_R^{\backslash d}} \mathcal{M}_{\boldsymbol{ir}}}.$$

*Proof.* The objective function of the many-body approximation is

$$L_{\mathrm{MBA}}(\mathcal{M}; \mathcal{Q}^{\mathrm{Tucker}}) = \sum_{\boldsymbol{i} \in \Omega_I} \sum_{\boldsymbol{r} \in \Omega_R} \mathcal{M}_{\boldsymbol{ir}} \log \mathcal{Q}_{\boldsymbol{ir}}^{\mathrm{Tucker}} \tag{16}$$

where

$$\mathcal{Q}_{i_1 \ldots i_D r_1 \ldots r_D}^{\mathrm{Tucker}} = \mathcal{G}_{r_1 \ldots r_D} A_{i_1 r_1}^{(1)} \ldots A_{i_D r_D}^{(D)}.$$

Since many-body approximation parameterizes tensors as discrete probability distributions, we optimize the above objective function with normalizing condition $\sum_{\boldsymbol{i} \in \Omega_I} \sum_{\boldsymbol{r} \in \Omega_R} \mathcal{Q}_{\boldsymbol{ir}}^{\mathrm{Tucker}} = 1$. Then, we consider the following Lagrange function:

$$\mathcal{L} = \sum_{\boldsymbol{i} \in \Omega_I} \sum_{\boldsymbol{r} \in \Omega_R} \mathcal{M}_{\boldsymbol{ir}} \log \mathcal{G}_{\boldsymbol{r}} A_{i_1 r_1}^{(1)} \ldots A_{i_D r_D}^{(D)} - \lambda \left( \sum_{\boldsymbol{i} \in \Omega_I} \sum_{\boldsymbol{r} \in \Omega_R} \mathcal{G}_{\boldsymbol{r}} A_{i_1 r_1}^{(1)} \ldots A_{i_D r_D}^{(D)} - 1 \right)$$

To reduce the scaling redundancy and decouple the normalizing condition, we introduce scaled factor matrices $\tilde{A}^{(d)}$ as

$$\tilde{A}_{i_d r_d}^{(d)} = \frac{A_{i_d r_d}^{(d)}}{a_{r_d}^{(d)}}, \quad \text{where } a_{r_d}^{(d)} = \sum_{i_d} A_{i_d r_d}, \tag{17}$$

and the scaled core tensor,

$$\tilde{\mathcal{G}}_{\boldsymbol{r}} = \mathcal{G}_{\boldsymbol{r}} a_{r_1}^{(1)} \ldots a_{r_D}^{(D)}.$$

The normalizing condition $\sum_{\boldsymbol{i} \in \Omega_I} \sum_{\boldsymbol{r} \in \Omega_R} \mathcal{Q}_{\boldsymbol{ir}}^{\mathrm{Tucker}} = 1$ guarantees the normalization of the core tensor $\tilde{\mathcal{G}}$ as

$$\sum_{\boldsymbol{r} \in \Omega_R} \tilde{\mathcal{G}}_{\boldsymbol{r}} = 1. \tag{18}$$

The tensor $\mathcal{Q}^{\mathrm{Tucker}}$ can be represented with the above introduced tensors as

$$\mathcal{Q}_{\boldsymbol{ir}}^{\mathrm{Tucker}} = \mathcal{G}_{\boldsymbol{r}} A_{i_1 r_1}^{(1)} \ldots A_{i_D r_D}^{(D)} = \tilde{\mathcal{G}}_{\boldsymbol{r}} \tilde{A}_{i_1 r_1}^{(1)} \ldots \tilde{A}_{i_D r_D}^{(D)}.$$

We optimize $\tilde{\mathcal{G}}$ and $\tilde{A}_{i_d r_d}^{(d)}$ instead of $\mathcal{G}$ and $A_{i_d r_d}^{(d)}$. Thus the Lagrange function can be written as

$$\mathcal{L} = \sum_{\boldsymbol{i} \in \Omega_I} \sum_{\boldsymbol{r} \in \Omega_R} \mathcal{M}_{\boldsymbol{ir}} \log \tilde{\mathcal{G}}_{\boldsymbol{r}} \tilde{A}_{i_1 r_1}^{(1)} \ldots \tilde{A}_{i_D r_D}^{(D)} - \lambda \left( \sum_{\boldsymbol{r}} \tilde{\mathcal{G}}_{\boldsymbol{r}} - 1 \right) - \sum_{d=1}^{D} \sum_{r_d} \lambda_{r_d}^{(d)} \left( \sum_{i_d} \tilde{A}_{i_d r_d}^{(d)} - 1 \right). \tag{19}$$

The condition

$$\frac{\partial \mathcal{L}}{\partial \tilde{\mathcal{G}}_{\boldsymbol{r}}} = \frac{\partial \mathcal{L}}{\partial \tilde{A}_{i_d r_d}^{(d)}} = 0$$

leads the optimal core tensor and factor matrices

$$\tilde{\mathcal{G}}_{\boldsymbol{r}} = \frac{1}{\lambda} \sum_{\boldsymbol{i} \in \Omega_I} \mathcal{M}_{\boldsymbol{ir}}, \quad \tilde{A}_{i_d r_d}^{(d)} = \frac{1}{\lambda_{r_d}^{(d)}} \sum_{\boldsymbol{i} \in \Omega_I^{\backslash d}} \sum_{\boldsymbol{r} \in \Omega_R^{\backslash d}} \mathcal{M}_{\boldsymbol{ir}}.$$

The values of Lagrange multipliers are identified by the normalizing conditions (17) and (18) as

$$\lambda = \sum_{\boldsymbol{i} \in \Omega_I} \sum_{\boldsymbol{r} \in \Omega_R} \mathcal{M}_{\boldsymbol{ir}}, \quad \lambda_{r_d}^{(d)} = \sum_{\boldsymbol{i} \in \Omega_I} \sum_{\boldsymbol{r} \in \Omega_R^{\backslash d}} \mathcal{M}_{\boldsymbol{ir}}.$$

$\square$

**Theorem 3** (The closed form of the optimal M-step in Train decomposition). *For a given tensor $\mathcal{M} \in \mathbb{R}_{\geq 0}^{I_1 \times \cdots \times I_D \times R_1 \times \cdots \times R_{D-1}}$, its many-body approximation with interactions described in Figure 3(c) is given as*

$$\mathcal{G}^{(d)}_{r_{d-1} i_d r_d} = \frac{\sum_{\boldsymbol{i} \in \Omega_I^{\backslash d}} \sum_{\boldsymbol{r} \in \Omega_R^{\backslash d, d-1}} \mathcal{M}_{\boldsymbol{ir}}}{\sum_{\boldsymbol{i} \in \Omega_I} \sum_{\boldsymbol{r} \in \Omega_R^{\backslash d}} \mathcal{M}_{\boldsymbol{ir}}}$$

*for $d = 1, \ldots, D$, assuming $r_0 = r_D = 1$.*

*Proof.* The objective function of the many-body approximation is

$$L_{\mathrm{MBA}}(\mathcal{M}; \mathcal{Q}^{\mathrm{Train}}) = \sum_{\boldsymbol{i} \in \Omega_I} \sum_{\boldsymbol{r} \in \Omega_R} \mathcal{M}_{\boldsymbol{ir}} \log \mathcal{Q}^{\mathrm{Train}}_{\boldsymbol{ir}}$$

where

$$\mathcal{Q}^{\mathrm{Train}}_{i_1 \ldots i_D r_1 \ldots r_D} = \mathcal{G}^{(1)}_{i_1 r_1} \mathcal{G}^{(2)}_{r_1 i_2 r_2} \ldots \mathcal{G}^{(D)}_{r_{D-1} i_D}.$$

Since many-body approximation parameterizes tensors as discrete probability distributions, we optimize the above objective function with normalizing condition $\sum_{\boldsymbol{i} \in \Omega_I} \sum_{\boldsymbol{r} \in \Omega_R} \mathcal{Q}^{\mathrm{Train}}_{\boldsymbol{ir}} = 1$. Then, we consider the following Lagrange function:

$$\mathcal{L} = \sum_{\boldsymbol{i} \in \Omega_I} \sum_{\boldsymbol{r} \in \Omega_R} \mathcal{M}_{\boldsymbol{ir}} \log \mathcal{G}^{(1)}_{i_1 r_1} \mathcal{G}^{(2)}_{r_1 i_2 r_2} \ldots \mathcal{G}^{(D)}_{r_{D-1} i_D} - \lambda \left( \sum_{\boldsymbol{i} \in \Omega_I} \sum_{\boldsymbol{r} \in \Omega_R} \mathcal{G}^{(1)}_{i_1 r_1} \mathcal{G}^{(2)}_{r_1 i_2 r_2} \ldots \mathcal{G}^{(D)}_{r_{D-1} i_D} - 1 \right)$$

To decouple the normalizing condition and make the problem simpler, we introduce scaled core tensors $\tilde{\mathcal{G}}^{(1)}, \ldots, \tilde{\mathcal{G}}^{(D-1)}$ that are normalized over $r_{d-1}$ and $i_d$ as

$$\tilde{\mathcal{G}}^{(d)}_{r_{d-1} i_d r_d} = \frac{g^{(d-1)}_{r_{d-1}}}{g^{(d)}_{r_d}} \mathcal{G}^{(d)}_{r_{d-1} i_d r_d},$$

where we define

$$g^{(d)}_{r_d} = \sum_{r_{d-1}} \sum_{i_d} \mathcal{G}^{(d)}_{r_{d-1} i_d r_d} g^{(d-1)}_{r_{d-1}},$$

with $g_{r_0} = 1$. We assume $r_0 = r_D = 1$. Using the scaled core tensors, the tensor $\mathcal{Q}^{\mathrm{Train}}$ can be written as

$$\mathcal{Q}^{\mathrm{Train}}_{i_1 \ldots i_D r_1 \ldots r_D} = \mathcal{G}^{(1)}_{i_1 r_1} \mathcal{G}^{(2)}_{r_1 i_2 r_2} \ldots \mathcal{G}^{(D)}_{r_{D-1} i_D} = \tilde{\mathcal{G}}^{(1)}_{i_1 r_1} \tilde{\mathcal{G}}^{(2)}_{r_1 i_2 r_2} \ldots \tilde{\mathcal{G}}^{(D)}_{r_{D-1} i_D}$$

with

$$\tilde{\mathcal{G}}^{(D)}_{r_{D-1} i_D} = \frac{1}{g^{(D-1)}_{r_{D-1}}} \mathcal{G}^{(D)}_{r_{D-1} i_D}. \tag{20}$$

The matrix $\tilde{\mathcal{G}}^{(D)}$ is normalized, satisfying $\sum_{r_{D-1}} \sum_{i_D} \tilde{\mathcal{G}}^{(D)}_{r_{D-1} i_D} = 1$. Thus, the Lagrange function can be written as

$$\mathcal{L} = \sum_{\boldsymbol{i} \in \Omega_I} \sum_{\boldsymbol{r} \in \Omega_R} \mathcal{M}_{\boldsymbol{ir}} \log \tilde{\mathcal{G}}^{(1)}_{i_1 r_1} \tilde{\mathcal{G}}^{(2)}_{r_1 i_2 r_2} \ldots \tilde{\mathcal{G}}^{(D)}_{r_{D-1} i_D} - \sum_{d=1}^{D-1} \lambda^{(d)}_{r_d} \left( \sum_{r_{d-1}} \sum_{i_d} \tilde{\mathcal{G}}^{(d)}_{r_{d-1} i_d r_d} - 1 \right)$$

$$- \lambda^{(D)} \left( \sum_{r_{D-1}} \sum_{i_d} \tilde{\mathcal{G}}^{(D)}_{r_{D-1} i_D} - 1 \right). \tag{21}$$

The critical condition

$$\frac{\partial \mathcal{L}}{\partial \tilde{\mathcal{G}}^{(d)}_{r_{d-1} i_d r_d}} = 0$$

leads the optimal core tensors

$$\tilde{\mathcal{G}}^{(d)}_{r_{d-1} i_d r_d} = \frac{1}{\lambda^{(d)}_{r_d}} \sum_{\boldsymbol{i} \in \Omega_I^{\backslash d}} \sum_{\boldsymbol{r} \in \Omega_R^{\backslash d, d-1}} \mathcal{M}_{\boldsymbol{ir}},$$

where the values of multipliers are identified by the normalizing conditions in Equation (20) as

$$\lambda^{(d)}_{r_d} = \sum_{\boldsymbol{i} \in \Omega_I} \sum_{\boldsymbol{r} \in \Omega_R^{\backslash d}} \mathcal{M}_{\boldsymbol{ir}}.$$

$\square$

## A.2 Proofs for EM-algorithm

We prove the propositions used to derive the EM algorithm for non-negative tensor mixture learning in Section 4. Furthermore, we show the convergence of the proposed method in Theorem 4.

**Proposition 1.** *For any tensors $\Phi^1, \ldots, \Phi^K$ that satisfies $\sum_{k=1}^{K} \sum_{\boldsymbol{r} \in \Omega_{R^k}} \Phi_{\boldsymbol{ir}}^k = 1$, the cross entropy $L(\hat{\mathcal{Q}})$ in Equation (7) can be bounded as follows:*

$$L(\hat{\mathcal{Q}}) \geq \overline{L}(\hat{\mathcal{Q}}, \Phi) = \sum_{\boldsymbol{i} \in \Omega_I} \sum_{k=1}^{K} \sum_{\boldsymbol{r} \in \Omega_{R^k}} \mathcal{T}_{\boldsymbol{i}} \Phi_{\boldsymbol{ir}}^k \log \frac{\hat{\mathcal{Q}}_{\boldsymbol{ir}}^k}{\Phi_{\boldsymbol{ir}}^k}.$$

*Proof.* For any tensors $\Phi^1, \ldots, \Phi^K$ that satisfies $\sum_{k=1}^{K} \sum_{\boldsymbol{r} \in \Omega_{R^k}} \Phi_{\boldsymbol{ir}}^k = 1$, we can transform the cross entropy $L(\hat{\mathcal{Q}})$ as follows:

$$
\begin{aligned}
L(\hat{\mathcal{Q}}) &= \sum_{\boldsymbol{i} \in \Omega_I} \mathcal{T}_{\boldsymbol{i}} \log \mathcal{P}_{\boldsymbol{i}} \\
&= \sum_{\boldsymbol{i} \in \Omega_I} \mathcal{T}_{\boldsymbol{i}} \log \sum_{\boldsymbol{r} \in \Omega_{R^k}} \hat{\mathcal{Q}}_{\boldsymbol{ir}}^k \\
&= \sum_{\boldsymbol{i} \in \Omega_I} \mathcal{T}_{\boldsymbol{i}} \log \sum_{k=1}^{K} \sum_{\boldsymbol{r} \in \Omega_{R^k}} \frac{\Phi_{\boldsymbol{ir}}^k \hat{\mathcal{Q}}_{\boldsymbol{ir}}^k}{\Phi_{\boldsymbol{ir}}^k} \\
&\geq \sum_{\boldsymbol{i} \in \Omega_I} \sum_{k=1}^{K} \sum_{\boldsymbol{r} \in \Omega_{R^k}} \mathcal{T}_{\boldsymbol{i}} \Phi_{\boldsymbol{ir}}^k \log \frac{\hat{\mathcal{Q}}_{\boldsymbol{ir}}^k}{\Phi_{\boldsymbol{ir}}^k} = \overline{L}(\mathcal{Q}, \Phi)
\end{aligned}
\tag{22}
$$

where the following relation, called the Jensen inequality (Jensen, 1906), is used:

$$f\left(\sum_{m=1}^{M} \lambda_m x_m\right) \geq \sum_{m=1}^{M} \lambda_m f(x_m) \tag{23}$$

for any concave function $f : \mathbb{R} \to \mathbb{R}$ and real numbers $\lambda_1, \ldots, \lambda_M$ that satisfies $\sum_{m=1}^{M} \lambda_m = 1$. The inequality is adaptable in Equation (22) because the logarithm function is a concave function. $\square$

**Proposition 2.** *In E-step, the optimal update for $\Phi$ is given as*

$$\tilde{\Phi}_{\boldsymbol{ir}}^k = \frac{\hat{\mathcal{Q}}_{\boldsymbol{ir}}^k}{\sum_{k=1}^{K} \sum_{\boldsymbol{r} \in \Omega_{R^k}} \hat{\mathcal{Q}}_{\boldsymbol{ir}}^k}.$$

*Proof.* We put Equation (10) into the lower bound $\overline{L}$ in Equation (8),

$$
\begin{aligned}
\overline{L}(\hat{\mathcal{Q}}, \tilde{\Phi}) &= \sum_{\boldsymbol{i} \in \Omega_I} \sum_{k=1}^{K} \sum_{\boldsymbol{r} \in \Omega_{R^k}} \mathcal{T}_{\boldsymbol{i}} \tilde{\Phi}_{\boldsymbol{ir}}^k \log \frac{\hat{\mathcal{Q}}_{\boldsymbol{ir}}^k}{\tilde{\Phi}_{\boldsymbol{ir}}^k} \\
&= \sum_{\boldsymbol{i} \in \Omega_I} \sum_{k=1}^{K} \sum_{\boldsymbol{r} \in \Omega_{R^k}} \mathcal{T}_{\boldsymbol{i}} \frac{\hat{\mathcal{Q}}_{\boldsymbol{ir}}^k}{\sum_{k=1}^{K} \sum_{\boldsymbol{r} \in \Omega_{R^k}} \hat{\mathcal{Q}}_{\boldsymbol{ir}}^k} \log \sum_{k=1}^{K} \sum_{\boldsymbol{r} \in \Omega_{R^k}} \hat{\mathcal{Q}}_{\boldsymbol{ir}}^k \\
&= \sum_{\boldsymbol{i} \in \Omega_I} \mathcal{T}_{\boldsymbol{i}} \log \sum_{k=1}^{K} \sum_{\boldsymbol{r} \in \Omega_{R^k}} \hat{\mathcal{Q}}_{\boldsymbol{ir}}^k \\
&= L(\hat{\mathcal{Q}}).
\end{aligned}
$$

Jensen's inequality in Equation (23) shows that

$$L(\hat{\mathcal{Q}}) = \overline{L}(\hat{\mathcal{Q}}, \tilde{\Phi}) \geq \overline{L}(\hat{\mathcal{Q}}, \Phi) \tag{24}$$

for any tensors $\Phi \in \mathcal{D}$ where $\mathcal{D} = \{ (\Phi^1, \ldots, \Phi^K) \mid \sum_k \sum_{\boldsymbol{r} \in \Omega_{R^k}} \Phi_{\boldsymbol{ir}}^k = 1 \}$. Thus, the tensors $\tilde{\Phi} = (\tilde{\Phi}^1, \ldots, \tilde{\Phi}^K)$ are optimal. $\square$

**Proposition 3.** *In M-step, the optimal update for the mixture ratio $\eta = (\eta^1, \ldots, \eta^K)$ that optimizes $J(\eta)$ in Equation (9) is given as*

$$\eta^k = \frac{\sum_{\boldsymbol{i}\in\Omega_I}\sum_{\boldsymbol{r}\in\Omega_{R^k}}\mathcal{T}_{\boldsymbol{i}}\Phi^k_{\boldsymbol{ir}}}{\sum_{\boldsymbol{i}\in\Omega_I}\sum_{k=1}^K\sum_{\boldsymbol{r}\in\Omega_{R^k}}\mathcal{T}_{\boldsymbol{i}}\Phi^k_{\boldsymbol{ir}}}$$

*Proof.* We optimize the decoupled objective function

$$J(\eta) = \sum_{\boldsymbol{i}\in\Omega_I}\sum_{k=1}^K\sum_{\boldsymbol{r}\in\Omega_{R^k}}\mathcal{T}_{\boldsymbol{i}}\Phi^k_{\boldsymbol{ir}}\log\eta^k$$

with conditions $\sum_{k=1}^K\eta^k = 1$ and $\eta^k \geq 0$. Thus we consider the following Lagrange function

$$\mathcal{L} = \sum_{\boldsymbol{i}\in\Omega_I}\sum_{k=1}^K\sum_{\boldsymbol{r}\in\Omega_{R^k}}\mathcal{T}_{\boldsymbol{i}}\Phi^k_{\boldsymbol{ir}}\log\eta^k - \lambda\left(\sum_{k=1}^K\eta^k - 1\right)$$

The condition $\partial\mathcal{L}/\partial\eta^k = 0$ leads to the optimal ratio

$$\eta^k = \frac{1}{\lambda}\sum_{\boldsymbol{i}\in\Omega_I}\sum_{\boldsymbol{r}\in\Omega_{R^k}}\mathcal{T}_{\boldsymbol{i}}\Phi^k_{\boldsymbol{ir}}$$

where the normalization identifies the multiplier $\lambda$ as

$$\lambda = \sum_{\boldsymbol{i}\in\Omega_I}\sum_{k=1}^K\sum_{\boldsymbol{r}\in\Omega_{R^k}}\mathcal{T}_{\boldsymbol{i}}\Phi^k_{\boldsymbol{ir}}.$$

$\square$

To the best of our knowledge, the general convergence theorem of the MU method is still an open problem, requiring proof of convergence for each minor change in the objective function, such as varying the low-rank structure, imposing symmetrical conditions, or adding a regularization term. On the other hand, the following Theorem 4 ensures that our framework converges regardless of such variations of the object function.

**Theorem 4.** *Mixture EM-tensor factorization always converges regardless of the choice of low-rank structure and mixtures.*

*Proof.* We prove the convergence of the EM algorithm from the fact that the objective function $L$ is bounded and that the E-step and M-step maximize the lower bound $\overline{L}$ with respect to $\Phi$ and $\hat{\mathcal{Q}}$, respectively. The E-step in iteration $t$ updates $\Phi_{t-1}$ by optimal $\tilde{\Phi}_t$ to maximize the lower bound for $\Phi$ such as

$$L(\hat{\mathcal{Q}}_t) = \overline{L}(\hat{\mathcal{Q}}_t, \tilde{\Phi}_t) \geq \overline{L}(\hat{\mathcal{Q}}_t, \Phi_{t-1}).$$

Then, the M-step in iteration $t$ updates $\hat{\mathcal{Q}}_t$ by $\hat{\mathcal{Q}}_{t+1}$ to maximize the lower bound for $\hat{\mathcal{Q}}$ such as

$$\overline{L}(\hat{\mathcal{Q}}_{t+1}, \tilde{\Phi}_t) \geq \overline{L}(\hat{\mathcal{Q}}_t, \tilde{\Phi}_t).$$

Again, the E-step in iteration $t+1$ updates $\Phi_t$ by $\tilde{\Phi}_{t+1}$ to maximize the lower bound for $\Phi$ such as

$$L(\hat{\mathcal{Q}}_{t+1}) \stackrel{\text{Eq.(24)}}{=} \overline{L}(\hat{\mathcal{Q}}_{t+1}, \tilde{\Phi}_{t+1}) \geq \overline{L}(\hat{\mathcal{Q}}_{t+1}, \Phi_t).$$

Combining the above three relations, we get

$$L(\hat{\mathcal{Q}}_{t+1}) = \overline{L}(\hat{\mathcal{Q}}_{t+1}, \tilde{\Phi}_{t+1}) \geq \overline{L}(\hat{\mathcal{Q}}_{t+1}, \tilde{\Phi}_t) \geq \overline{L}(\hat{\mathcal{Q}}_t, \tilde{\Phi}_t) = L(\hat{\mathcal{Q}}_t).$$

Thus, it holds that $L(\hat{\mathcal{Q}}_{t+1}) \geq L(\hat{\mathcal{Q}}_t)$. The algorithm converges because the cost function is bounded and would not be decreasing in each iteration. $\square$

# B ADDITONAL REMARKS

## B.1 TECHNICAL DETAIL FOR COMPLICATED LOW-RANK STRUCTURES

We decoupled the tensor many-body decomposition into independent problems in Section 4.3 using the basic property of logarithmic functions. We show here that the normalization for the model is satisfied when we use the solution formula of the many-body approximation for each decoupled problem.

In the following, we discuss the decomposition as described in Figure 4 as an example, while the generalization to arbitrary tree low-rank structures is straightforward. When we decouple the many-body approximation in Equation (15) into the many-body approximation corresponding to the Tucker and Train decompositions, we need to guarantee that the normalizing condition

$$\sum_{\boldsymbol{i} \in \Omega_I} \sum_{\boldsymbol{r} \in \Omega_R} \mathcal{Q}_{\boldsymbol{ir}} = 1 \tag{25}$$

is satisfied where we define

$$\mathcal{Q}_{\boldsymbol{ir}} = \mathcal{G}_{r_1 r_2 r_5} A_{i_1 r_1} B_{i_2 r_2} C_{r_5 r_6} D_{i_3 r_3} \mathcal{H}_{r_3 r_6 r_4} E_{r_4 i_4}. \tag{26}$$

As explained at the beginning of Section A.1, we reduce scaling redundancy by scaling each factor and decouple the Lagrange function into independent parts. More specifically, we define a single root tensor and introduce normalized factors that sums over the edges that lie below from the root. Although the choice of the root tensor is not unique, we let tensor $\mathcal{G}$ be the root tensor and introduce

$$\tilde{A}_{i_1 r_1} = \frac{1}{a_{r_1}} A_{i_1 r_1}, \quad \tilde{B}_{i_2 r_2} = \frac{1}{b_{r_2}} B_{i_2 r_2}, \quad \tilde{C}_{r_5 r_6} = \frac{h_{r_6}}{c_{r_5}} C_{r_5 r_6}, \tag{27}$$

$$\tilde{D}_{i_3 r_3} = \frac{1}{d_{r_3}} D_{i_3 r_3}, \quad \tilde{E}_{i_4 r_4} = \frac{1}{e_{r_4}} E_{i_4 r_4}, \quad \tilde{\mathcal{H}}_{r_3 r_4 r_6} = \frac{d_{r_3} e_{r_4}}{h_{r_6}} \mathcal{H}_{r_3 r_4 r_6} \tag{28}$$

where each normalizer is defined as

$$a_{r_1} = \sum_{i_1} A_{i_1 r_1}, \quad b_{r_2} = \sum_{i_2} B_{i_2 r_2}, \quad c_{r_5} = \sum_{r_6} C_{r_5 r_6} h_{r_6},$$

$$d_{r_3} = \sum_{i_3} D_{i_3 r_3}, \quad e_{r_4} = \sum_{i_4} E_{i_4 r_4}, \quad h_{r_6} = \sum_{r_3 r_4} d_{r_3} e_{r_4} \mathcal{H}_{r_3 r_4 r_6},$$

then it holds that

$$\sum_{i_1} \tilde{A}_{i_1 r_1} = \sum_{i_2} \tilde{B}_{i_2 r_2} = \sum_{r_6} \tilde{C}_{r_5 r_6} = \sum_{i_3} \tilde{D}_{i_3 r_3} = \sum_{i_4} \tilde{E}_{i_4 r_4} = \sum_{r_3 r_4} \tilde{\mathcal{H}}_{r_3 r_4 r_6} = 1.$$

We define the tensor $\tilde{\mathcal{G}}$ as $\tilde{\mathcal{G}}_{r_1 r_2 r_5} = a_{r_1} b_{r_2} c_{r_5} \mathcal{G}_{r_1 r_2 r_5}$ and putting Equations (27) and (28) into Equations (26) and (25), we obtain the normalizing condition for the root tensor $\mathcal{G}$ as

$$\sum_{r_1 r_2 r_5} \tilde{\mathcal{G}}_{r_1 r_2 r_5} = 1.$$

Then, the tensor $\mathcal{Q}$ can be written as

$$\mathcal{Q}_{\boldsymbol{ir}} = \tilde{\mathcal{G}}_{r_1 r_2 r_5} \tilde{A}_{i_1 r_1} \tilde{B}_{i_2 r_2} \tilde{C}_{r_5 r_6} \tilde{D}_{i_3 r_3} \tilde{\mathcal{H}}_{r_3 r_6 r_4} \tilde{E}_{r_4 i_4}. \tag{29}$$

The above approach to reduce scaling redundancy is illustrated in Figure 6. Finally, the original optimization problem with the Lagrange function

$$\mathcal{L} = \sum_{\boldsymbol{i} \in \Omega_I} \sum_{\boldsymbol{r} \in \Omega_R} \mathcal{M}_{\boldsymbol{ir}} \log \mathcal{Q}_{\boldsymbol{ir}} - \lambda \left( 1 - \sum_{\boldsymbol{i} \in \Omega_I} \sum_{\boldsymbol{r} \in \Omega_R} \mathcal{Q}_{\boldsymbol{ir}} \right)$$

is equivalent to the problem with the Lagrange function

$$\mathcal{L} = \mathcal{L}^{\text{Tucker}} + \mathcal{L}^{\text{Train}}$$

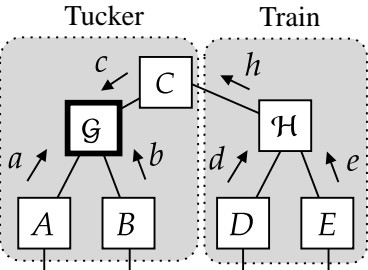

Figure 6: We normalize all tensors except for the root tensor, which is enclosed in a bold line. We then push the normalizer of each tensor, $a, b, c, d, e$, and $h$ on the root tensor. The root tensor absorbs scaling redundancy. This procedure decouples the Lagrangian $\mathcal{L}$ into two independent problems, $\mathcal{L}^{\text{Tucker}}$ and $\mathcal{L}^{\text{Train}}$

where

$$\mathcal{L}^{\text{Tucker}} = \sum_{\boldsymbol{i} \in \Omega_I} \sum_{\boldsymbol{r} \in \Omega_R} \mathcal{M}_{\boldsymbol{ir}} \log \tilde{G}_{r_1 r_2 r_5} \tilde{A}_{i_1 r_1} \tilde{B}_{i_2 r_2} \tilde{C}_{r_5 r_6} + \lambda^{\mathcal{G}} \left( \sum_{r_1 r_2 r_5} \tilde{\mathcal{G}}_{r_1 r_2 r_5} - 1 \right)$$

$$+ \sum_{r_1} \lambda_{r_1}^A \left( \sum_{i_1} \tilde{A}_{i_1 r_1} - 1 \right) + \sum_{r_2} \lambda_{r_2}^B \left( \sum_{i_2} \tilde{B}_{i_2 r_2} - 1 \right) + \sum_{r_5} \lambda_{r_5}^C \left( \sum_{r_6} \tilde{C}_{r_5 r_6} - 1 \right),$$

which is equivalent to the Lagrange function for the Tucker decomposition given in Equation (19) and

$$\mathcal{L}^{\text{Train}} = \sum_{\boldsymbol{i} \in \Omega_I} \sum_{\boldsymbol{r} \in \Omega_R} \mathcal{M}_{\boldsymbol{ir}} \log \tilde{\mathcal{H}}_{r_3 r_4 r_6} \tilde{D}_{i_3 r_3} \tilde{E}_{i_4 r_4}$$

$$+ \sum_{r_3} \lambda_{r_3}^D \left( \sum_{i_3} \tilde{D}_{i_3 r_3} - 1 \right) + \sum_{r_4} \lambda_{r_4}^E \left( \sum_{i_4} \tilde{E}_{i_4 r_4} - 1 \right) + \sum_{r_6} \lambda_{r_6}^{\mathcal{H}} \left( \sum_{r_3 r_4} \tilde{\mathcal{H}}_{r_3 r_4 r_6} - 1 \right),$$

which is also equivalent to the Lagrange function for the Train decomposition given in Equation (21) assuming $G^{(D)}$ is a normalized uniform tensor. For simplicity, we define tensors

$$\mathcal{M}_{i_1 i_2 r_1 r_2 r_5}^{\text{Tucker}} = \sum_{i_3 i_4} \sum_{r_3 r_4 r_6} \mathcal{M}_{\boldsymbol{ir}}, \quad \mathcal{M}_{i_3 i_4 r_3 r_4 r_6}^{\text{Train}} = \sum_{i_1 i_2} \sum_{r_1 r_2 r_5} \mathcal{M}_{\boldsymbol{ir}},$$

then, solve these independent many-body approximations by the closed-form solution by Equations (4) and (5) for given tensors $\mathcal{M}^{\text{Tucker}}$ and $\mathcal{M}^{\text{Train}}$, respectively, and multiply solutions to get optimal tensor $\mathcal{Q}$ as Equation (29), which satisfied the normalizing condition in Equation (25).

## B.2 EM TENSOR FACTORIZATION FOR GENERAL NON-NEGATIVE TENSORS

Non-negative tensor factorization optimizing the KL divergence is frequently used beyond density estimation and in various fields such as sound source separation (Kırbız & Günsel, 2014), computer vision (Kim et al., 2008; Phan & Cichocki, 2008), and data mining (Chi & Kolda, 2012; Takeuchi et al., 2013; Krompaß et al., 2013; Ermiş et al., 2015). Although the given tensor $\mathcal{T}$ is not necessarily normalized in such applications, the proposed framework can be used for them as follows. First, we obtain the total sum of the input tensor $\mu = \sum_{\boldsymbol{i}} \mathcal{T}_{\boldsymbol{i}}$ and then perform the factorization on the normalized tensor $\mathcal{T}$ by dividing all elements of $\mathcal{T}$ by $\mu$. Finally, all elements of the resulting tensor $\mathcal{P}$ are multiplied by $\mu$. This procedure is justified by the property of the KL divergence, $D_{KL}(\mu\mathcal{P}, \mu\mathcal{T}) = \mu D_{KL}(\mathcal{P}, \mathcal{T})$, where $\mu$ is any positive value.

## B.3 DEFINITION OF NORMALIZED MUTUAL INFORMATION

This section describes the definition of the mutual information (Strehl & Ghosh, 2002) used in the paper. For a given normalized non-negative tensor $\mathcal{T} \in \mathbb{R}_{\geq 0}^{I_1 \times \cdots \times I_D}$, we define its normalized mutual

---

**Algorithm 2:** EM Train decomposition

> **input** : Non-negative tensor $\mathcal{T}$ and train rank $R^{\text{Train}} = (R_1, \ldots, R_{D-1})$.
> Initialize $\mathcal{Q}^{\text{Train}}$;
> **repeat**
> $\quad$ $\mathcal{M}_{ir}^{\text{Train}} \leftarrow \mathcal{T}_i \mathcal{Q}_{ir}^{\text{Train}} / \mathcal{P}_i^{\text{Train}}$ for $i \in \Omega_I^{\text{o}}$; $\qquad\qquad\qquad\qquad$ // E-step
> $\quad$ **for** $d \leftarrow 1$ **to** $D$ **do**
> $\quad\quad$ $\lfloor$ Obtain $\mathcal{G}^{(\to d)}$ and $\mathcal{G}^{(D-d\leftarrow)}$ using Equation (13);
> $\quad$ Update $\mathcal{G}^{(1)}, \ldots, \mathcal{G}^{(D)}$ using Equation (14);
> $\quad$ $\mathcal{P}_i^{\text{Train}} \leftarrow \mathcal{G}_i^{(\to D)}$ for $i \in \Omega_I^{\text{o}}$;
> **until** *Convergence*;
> **return** $\mathcal{G}^{(1)}, \ldots, \mathcal{G}^{(D)}$

---

information between two modes $d, l \in [D]$ as

$$\mathrm{H}_{d,l} = \frac{\mathrm{I}_{d,l}}{\sqrt{\mathrm{H}_d \mathrm{H}_l}}$$

where mutual information $\mathrm{I} \in \mathbb{R}^{D \times D}$ and individual entropy $\mathrm{H} \in \mathbb{R}^D$ are defined as

$$\mathrm{I}_{d,l} = \sum_{i_d, i_l} \mathcal{T}_{i_d i_l}^{(d,l)} \log \frac{\mathcal{T}_{i_d i_l}^{(d,l)}}{\mathcal{T}_{i_d}^{(d)} \mathcal{T}_{i_l}^{(l)}}, \quad \mathrm{H}_d = \sum_{i_d} \mathcal{T}_{i_d}^{(d)} \log \mathcal{T}_{i_d}^{(d)}$$

for marginalized tensors

$$\mathcal{T}_{i_d i_l}^{(d,l)} = \sum_{i \in \Omega_I^{\backslash d,l}} \mathcal{T}_i, \quad \mathcal{T}_{i_d}^{(d)} = \sum_{i \in \Omega_I^{\backslash d}} \mathcal{T}_i.$$

### B.4 NOTATION, TERMINOLOGY, AND EVALUATION METRIC

Although all symbols and technical terms are properly introduced in the main text, we provide our notation here for readability.

The symbol $\mathbb{R}_{\geq 0}$ denotes the set of non-negative real numbers. The set of all natural numbers less than or equal to a natural number $K$ is denoted by $[K]$. We use the Landau symbol $O$ for computational time complexity.

**Tensors** We refer to a tensor whose elements all are nonnegative as a nonnegative tensor. The axes of a tensor are called its modes. The number of modes is called the order. For example, a vector is a first-order tensor, and a matrix is a second-order tensor. Tensors are denoted by calligraphic capital letters, as $\mathcal{T}, \mathcal{P}, \mathcal{Q}, \mathcal{M}$. A non-negative tensor whose sum over all indices is 1 is called a normalized tensor. Although the beginning and the last core tensors of the tensor-train format are matrices rather than tensors, they are denoted by the calligraphic letters $\mathcal{G}^{(1)}$ and $\mathcal{G}^{(D)}$ for notational convenience. The element-wise product of a normalized non-negative tensor $\mathcal{Q}^k$ and a weight $\eta^k$ is written with a hat, e.g., $\hat{\mathcal{Q}}^k = \eta^k \mathcal{Q}^k$. In the supplementary materials, tensors and matrices whose sums over some indices equal 1 are marked with a tilde as $\tilde{\mathcal{Q}}$.

**EM algorithm** The solution spaces for the E and M steps are represented by $\mathcal{D}$ and $\mathcal{B}$, respectively. Since the solution space of the M-step is equivalent to that of the many-body approximation for tensors, we use $\mathcal{B}$ to denote both. The objective function of the EM algorithm is $L$, and its lower bound is written using overlined $L$ as $\overline{L}$.

**Indices** The visible variables $i = (i_1, \ldots, i_D)$ and hidden variables $r = (r_1, \ldots, r_V)$ are denoted as lower subscripts. The index set of visible variables and hidden variables are denoted as $\Omega_I$ and $\Omega_R$, respectively. Multiple tensors are represented using superscripts with parentheses, such as $\mathcal{G}^{(1)}, \ldots, \mathcal{G}^{(D)}$. Indices for mixtures are expressed as superscripts without parentheses. For example, the mixing ratio is represented by $\eta = (\eta^1, \ldots, \eta^K)$. The symbol $\Omega_I^{\text{o}}$ is for the set of indices of nonzero values of the tensor $\mathcal{T}$, that is, $\Omega_I^{\text{o}} = \{ i \mid \mathcal{T}_i \neq 0 \} \subseteq \Omega_I$.

Table 3: List of baselines

|  | Library / Paper | Function | Loss | Non-negativity |
|---|---|---|---|---|
| CNMFOPT | Ibrahim & Fu (2021) | - | KL divergence | Yes |
| MPS | Glasser et al. (2019) | PositiveMPS | KL divergence | Yes |
| BM | Glasser et al. (2019) | RealBorn | KL divergence | Yes |
| LPS | Glasser et al. (2019) | RealLPS | KL divergence | Yes |
| KLNTDMU | nn_fac | ntd_mu | KL divergence | Yes |
| KLCPMU | nn_fac | ntf_mu | KL divergence | Yes |
| EMCP | — | — | KL divergence | Yes |
| CP | Tensorly | parafac | Frobenius norm | No |
| NNCP | Tensorly | non_negative_parafac | Frobenius norm | Yes |
| NNCPHALS | Tensorly | non_negative_parafac_hals | Frobenius norm | Yes |
| Tucker | Tensorly | tucker | Frobenius norm | No |
| NNTucker | Tensorly | non_negative_tucker | Frobenius norm | Yes |
| NNTuckerHALS | Tensorly | non_negative_tucker_hals | Frobenius norm | Yes |
| Train | Tensorly | tensor_train | Frobenius norm | No |

**Metric** The proposed framework and some baseline methods, MPS, BM, LPS, KLCPMU, and KLNTDMU optimize the KL divergence between tensors (Yang et al., 2011), which is defined as

$$D_{KL}(\mathcal{T}, \mathcal{P}) = \sum_{\boldsymbol{i} \in \Omega_I} \left\{ \mathcal{T}_{\boldsymbol{i}} \log \frac{\mathcal{T}_{\boldsymbol{i}}}{\mathcal{P}_{\boldsymbol{i}}} - \mathcal{T}_{\boldsymbol{i}} + \mathcal{P}_{\boldsymbol{i}} \right\},$$

where $\mathcal{T}$ and $\mathcal{P}$ can be non-normalized or normalized tensors. For a given tensor $\mathcal{T}$, the optimization of the KL divergence for the tensor $\mathcal{P}$ is equivalent to the maximization of the cross-entropy or minimizing the negative log-likelihood per sample. Other baseline methods, CP, Tucker, TT, NNCP, NNCPHALS, NNTucker, and NNTuckerHALS optimize the Frobenius norm, which is defined as

$$\|\mathcal{T} - \mathcal{P}\|_F = \sqrt{\sum_{\boldsymbol{i} \in \Omega_I} (\mathcal{T}_{\boldsymbol{i}} - \mathcal{P}_{\boldsymbol{i}})},$$

for given tensor $\mathcal{T}$. Since our motivation is density estimation, we evaluate each method with the negative log-likelihood, regardless of which objective function is used in the optimization.

## B.5 LIMITATIONS

The proposed framework only works on non-negative tensors. The theoretical analysis supporting the generalization performance of the mixture models remains in future work. While this study discussed low-rank structures with tree structures, such as CP, Tucker, and Train decomposition, and their combinations, we did not discuss tensor networks with loops. We have empirically examined the effectiveness of the proposed method only for discrete density estimation. The method cannot be used directly in situations where there are missing values in the data. The number of ranks that need to be tuned is larger in mixture models than in the non-mixture low-rank model. The method to decouple a complicated low-rank structure into solvable CP, Tucker, and Train decompositions is not unique. Thus, establishing an efficient decoupling method is also a subject for future work.

## C ADDITIONAL EXPERIMENTAL RESULTS

In this section, we discuss additional experimental results that could not be included in the main text due to page limitations. For convenience, we summarized the baseline methods in Table 3. We note that some baselines do not guarantee non-negativity or normalization, and thus, heuristics are needed as described in Section D.2.2.

The last part of this section is organized as follows. In Section C.1, we perform experiments on synthetic data to verify that the proposed algorithm can estimate the true distribution. In Section C.2, we observe that the adaptive noise term stabilizes the learning and significantly improves the validation

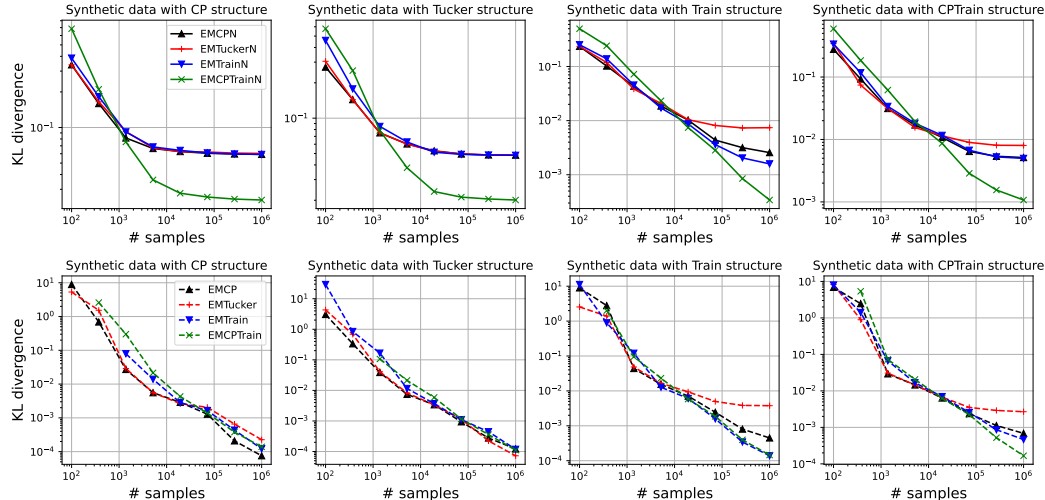

Figure 7: The KL divergence from the true distribution to the reconstructed tensor under varying numbers of available samples. The solid line represents the low-rank model with an adaptive noise term (top), while the dashed line represents the low-rank model without an adaptive noise term (bottom).

error. In Sections C.3, C.4, and C.5, we see the generalized performance comparing different optimization methods for the same low-rank model while we compared the performance between different models in Table 1 in the main text. In Section C.6, we verify the number of EM iterations required for convergence since it is often pointed out that the EM algorithm requires a lot of iterations to converge (Ng et al., 2012; Chege et al., 2022).

## C.1 SYNTHETIC-DATA SIMULATIONS

Since the true distribution is not given in the experiments using real data, it is not possible to verify whether the proposed algorithm can estimate the true distribution. Therefore, we perform experiments on synthetic data with given true distributions and verify that the proposed algorithm can estimate the true distribution if the number of samples is sufficient.

We synthesize $8 \times 8 \times 8 \times 8$ non-negative normalized tensors $\mathcal{U}^{\mathrm{CP}}$, $\mathcal{U}^{\mathrm{Tucker}}$, and $\mathcal{U}^{\mathrm{Train}}$ as follows:

$$\mathcal{U}^{\mathrm{CP}}_{i_1 i_2 i_3 i_4} = \sum_r \mathcal{Q}^{\mathrm{CP}}_{i_1 i_2 i_3 i_4 r}, \quad \mathcal{U}^{\mathrm{Tucker}}_{i_1 i_2 i_3 i_4} = \sum_{r_1 r_2 r_3} \mathcal{Q}^{\mathrm{Tucker}}_{i_1 i_2 i_3 i_4 r_1 r_2 r_3 r_4}, \quad \mathcal{U}^{\mathrm{Train}}_{i_1 i_2 i_3 i_4} = \sum_{r_1 r_2 r_3} \mathcal{Q}^{\mathrm{Train}}_{i_1 i_2 i_3 i_4 r_1 r_2 r_3}.$$

Each element in the factors of the above tensors is independently sampled from a normal distribution, and its absolute value is taken. We also define the mixture of $\mathcal{U}^{\mathrm{CP}}$ and $\mathcal{U}^{\mathrm{Train}}$ as

$$\mathcal{U}^{\mathrm{CPTrain}} = \frac{1}{2}\mathcal{U}^{\mathrm{CP}} + \frac{1}{2}\mathcal{U}^{\mathrm{Train}}.$$

The tensor rank of $\mathcal{U}^{\mathrm{CP}}$ is 8, $\mathcal{U}^{\mathrm{Tucker}}$ is $(3, 3, 3, 3)$, and $\mathcal{U}^{\mathrm{Train}}$ is $(4, 4, 4)$. We add a noise term with the weight $\eta^{\mathrm{noise}} = 0.10$ to these synthesized tensors and then normalize them. These tensors are regarded as true distributions. We obtain the samples from $\mathcal{U}^{\mathrm{CP}}, \mathcal{U}^{\mathrm{Tucker}}, \mathcal{U}^{\mathrm{Train}}$ and $\mathcal{U}^{\mathrm{CPTrain}}$ and randomly divide them into training and validation data.

We construct training tensors $\mathcal{T}$ from training data and factorize them using the proposed methods to obtain reconstructed low-rank tensors $\mathcal{P}^k$ for $k \in \{\mathrm{CP}, \mathrm{Tucker}, \mathrm{Train}, \mathrm{CPTrain}\}$. Next, we estimate the tensor ranks that best fit the validation data. Finally, we evaluate the KL divergence from the true distribution $\mathcal{U}^k$ to the reconstruction $\mathcal{P}^k$. This process is repeated with varying sample sizes, and the results are shown in Figure 7. The results show that the proposed method can estimate the true distribution more accurately as the number of samples is increased.

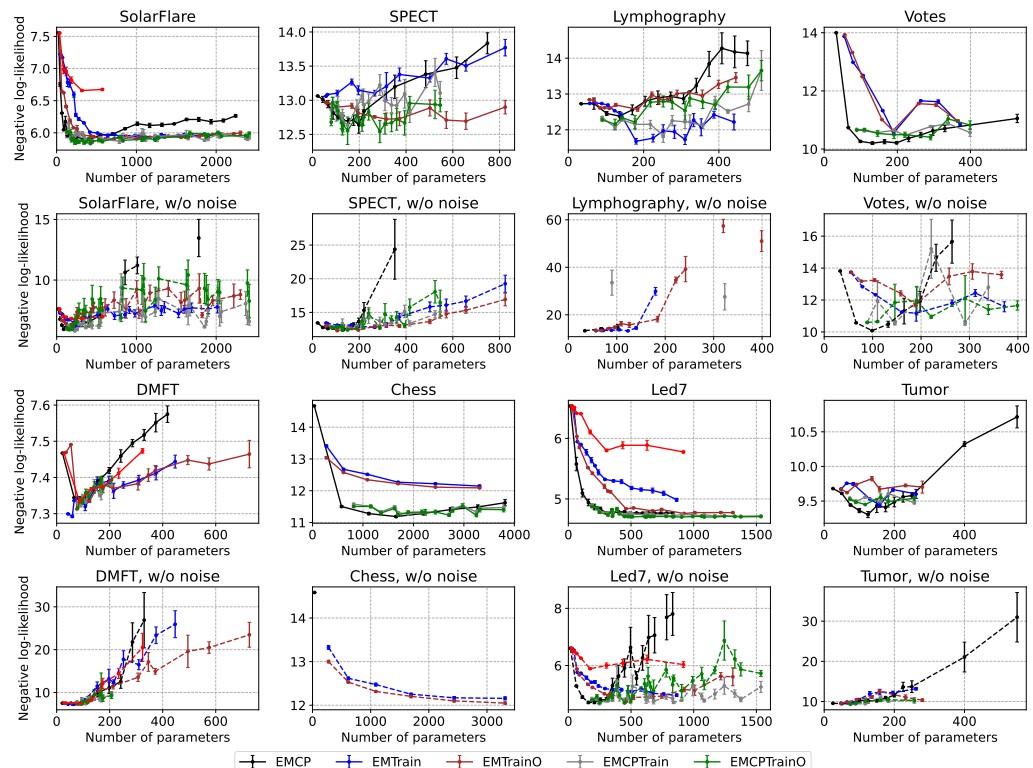

Figure 8: Validation error for each number of parameters for each dataset with adaptive noise learning (solid line) and without adaptive noise learning (dashed line).

## C.2 EFFECT OF THE ADAPTIVE NOISE TERM

We compare the validation error of the proposed model with and without the adaptive noise term in Figure 8. Focusing on the vertical axis, we can see that the models not including the adaptive noise term had significantly larger errors when the models are specified with large numbers of parameters and thus prone to overfitting.

## C.3 DIRECT COMPARISON AMONG MPS, EM-TRAIN, AND LSTRAIN

Both the proposed EMTrain and the baseline MPS are the same model, with different optimization methods. Although the ranks in both models are vector values, the official implementation of the MPS [2] assumes that each element of the rank is the same while the rank of proposed methods has been tuned as described in Section D.2. Therefore, for a direct and more fair comparison, we have conducted the experiment again fixing the range of train ranks for EMTrainN and EMTrainON to be $(r, ..., r)$ for $r = 1, ..., 8$. Furthermore, we compared the results with conventional tensor-train decompositions (Oseledets, 2011) that optimize the Frobenius norm (LSTrain). The results are provided in Table 4 where we observe that EMTrainON outperforms MPS and LSTrain for all datasets. Chess data could not be decomposed by LSTrain. This is because NumPy does not support dense tensors with larger than 33 dimensions while LSTrain needs to treat the data as a dense format to perform high-order SVD.

## C.4 COMPARISON WITH EXISTING TUCKER METHODS

We provide in Table 5 the results of our experiments comparing with Tucker (Tucker, 1966), NNTucker (Kim & Choi, 2007), NNTuckerHALS (Phan & Cichocki, 2011), and NTDMU (Kim et al., 2008) to validate the usefulness of the proposed methods, EMTucker and EMTuckerN. Because the Tucker structure requires a large memory requirement to store the dense core tensor, we could

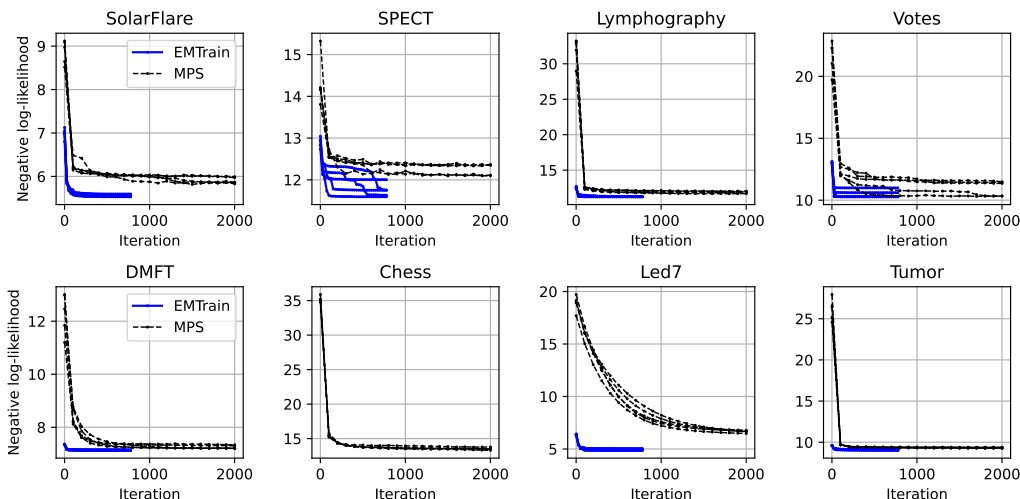

Figure 9: Loss curves for each dataset trained by the batch gradient method (MPS) and the proposed method (EMTrain).

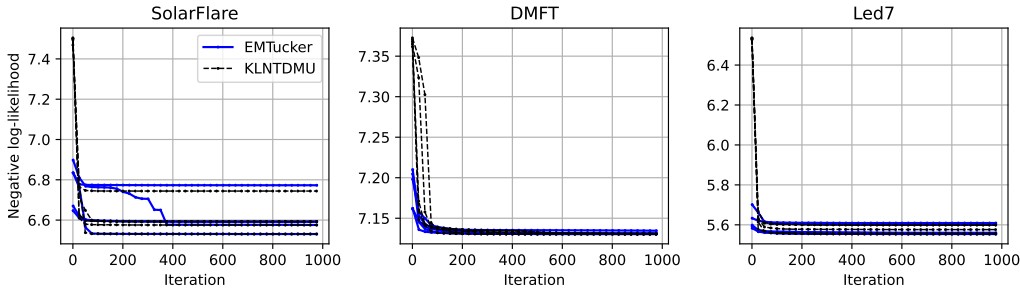

Figure 10: Loss curves for each dataset trained by the multiplicative update (KLNTDMU) and the proposed method (EMTucker).

perform the experiments on only three datasets with relatively small orders, SolarFlare, DMFT, and Led7. We describe the rank tuning and convergence conditions in Section D.2.2. The adaptive noise term makes EMTuckerN more stable than EMTucker. Since Tucker, NNTucker, and NNTuckerHALS optimize the Frobenius norm, we observe that the negative log-likelihood is relatively large for these methods.

## C.5 COMPARISON WITH EXISTING CPD METHODS

We also provide in Table 6 the results of our experiments comparing with CP (Kolda & Bader, 2009), NNCP (Shashua & Hazan, 2005), NNCPHALS (Cichocki & Phan, 2009), and KLCPMU (Cichocki et al., 2009) to validate the usefulness of the proposed method, EMCPN. We observed that EMCPN is more stable than EMCP due to the adaptive noise term. While KLCPMU performed the best generalization on SolarFlare and DMFT, it was not applicable to Chess and Votes, which have a large number of random variables. Specifically, KLCPMU did not converge for Votes even 72 hours after the experiment started. These baselines could not handle chess because Numpy cannot handle tensors of more than 33 dimensions.

## C.6 CONVERGENCE SPEED OF EM-BASED ALGORITHM

We additionally performed experiments to investigate the difference in convergence performance between the proposed and existing methods that have the same objective function and low-rank

Table 4: Negative log likelihood per test samples

|  | MPS | LSTrain | EMTrainN | EMTrainON |
|---|---|---|---|---|
| SolarFlare | 6.08(0.05) | 6.23(0.00) | **6.02**(0.04) | 6.07(0.06) |
| SPECT | 14.83(2.05) | 14.00(0.00) | 12.08(0.00) | **11.22**(0.11) |
| Lymphography | 13.03(0.04) | 17.24(0.00) | 12.69(0.19) | **12.10**(0.12) |
| Votes | 11.80(0.26) | 12.25(0.00) | **10.31**(0.04) | 10.56(0.16) |
| Tumor | 9.54(0.19) | 11.31(0.00) | **9.23**(0.11) | 9.52(0.00) |
| Chess | 12.45(0.17) | — | **12.07**(0.02) | 12.07(0.05) |
| Led7 | 5.86(0.17) | 5.15(0.00) | 5.11(0.08) | **4.82**(0.01) |
| DMFT | 7.25(0.05) | 7.42(0.00) | **7.22**(0.00) | 7.26(0.00) |

Table 5: Negative log likelihood per test samples

|  | Tucker | NNTucker | NNTuckerHALS | NTDMU | EMTucker | EMTuckerN |
|---|---|---|---|---|---|---|
| SolarFlare | 10.86(0.39) | 11.85(1.60) | 10.65(0.43) | 6.69(0.00) | 6.78(0.15) | **6.60**(0.04) |
| DMFT | 7.34(0.00) | 7.29(0.06) | 7.32(0.00) | **7.22**(0.00) | 7.31(0.12) | 7.24(0.02) |
| Led7 | 6.37(0.00) | 7.61(0.48) | 6.45(0.11) | 6.25(0.00) | 6.04(0.11) | **5.86(0.03)** |

structures but different optimization techniques. For a fair comparison, we did not include the adaptive noise term in the proposed methods in the following experiments.

**EMTrain and MPS**   We compare the convergence of the proposed EMTrain and the batch-gradient-based MPS (Glasser et al., 2019), which are the equivalent models using different optimizations. The computation complexity per iteration of the EMTrain and MPS is $O(IR^2D)$ and $O(DBNR^2)$, respectively where $I$ is the degrees of freedom of the variables, $D$ is the number of discrete variables, $(R, \ldots, R)$ is the train-rank, $B$ is the batch size, and $N$ is the number of observed samples. We chose the ranks and learning rates at which the MPS minimizes the validation error. The batch size for MPS follows the description in Section D.2. The results in Figure 9 imply that the proposed method converges more rapidly than the baseline method. We also observed a stable curve of the proposed method, given the simultaneous updating of all parameters and the monotonically decreasing nature of the objective function. Since the proposed method EMTrain has no adaptive noise term, the optimization was unstable for the sparsest Chess dataset with NaN value in the objective function. In this experiment, the initial values of the EMTrain were defined in the same way as for MPS. In particular, each element of the core tensor was sampled from a standard distribution and then squared.

**EMTucker and KLNTDMU**   In addition, we compare EMTucker and KLNTDMU, which are also equivalent models using different optimizations. We chose the learning rates at which the KLNTDMU minimizes the validation error. The results in Figure 10 imply that the proposed EMTucker have comparable convergence performance with the multiplicative update-based methods. In this experiment, the initial values of the EMTucker were defined in the same way as for KLNTDMU.

# D   EXPERIMENTAL DETAILS

## D.1   EXPERIMENTAL SETUP

We download four categorical tabular datasets SolarFlare, SPECT, Lymphography, and Chess from the UCI database,[1] none of which contain any missing values, and two preprocessed categorical tabular datasets Votes and Tumor from the official repository of baselines (Glasser et al., 2019). All these datasets, except Chess, are used in the paper of the baseline methods (Glasser et al., 2019). We also download two categorical tabular datasets, Led7 and DMFT, from Penn Machine Learning Benchmarks (Olson et al., 2017). Each dataset contains $N$ categorical samples $x_n = (i_1, ..., i_D)$ for

---

[1]https://archive.ics.uci.edu/,

Table 6: Negative log likelihood per test samples

|            | CP          | KLCPMU        | NNCP         | NNCPHALS     | EMCP         | EMCPN         |
|------------|-------------|---------------|--------------|--------------|--------------|---------------|
| SolarFlare | 6.18(0.01)  | **5.89**(0.03)| 6.29(0.07)   | 6.25(0.06)   | 6.04(0.19)   | 5.95(0.02)    |
| SPECT      | inf(nan)    | **11.24**(0.00)| 14.22(0.44) | 14.22(0.11)  | 11.28(0.56)  | 11.46(0.19)   |
| Lympho.    | inf(nan)    | 13.02(0.00)   | 17.13(0.19)  | 17.13(0.19)  | 13.02(0.00)  | **12.58**(0.00)|
| Votes      | 12.33(0.36) | 10.56(0.27)   | 12.10(0.04)  | 12.08(0.02)  | 10.83(0.43)  | **10.34**(0.07)|
| Tumor      | 11.23(0.28) | 9.30(0.40)    | 11.39(0.27)  | 11.24(0.42)  | **9.11**(0.22)| 9.21(0.06)   |
| Chess      | —           | —             | —            | —            | inf(nan)     | **11.15**(0.11)|
| Led7       | 4.75(0.03)  | 4.76(0.07)    | **4.70**(0.01)| 4.71(0.02)  | 4.75(0.07)   | 4.82(0.00)    |
| DMFT       | 7.23(0.11)  | **7.12**(0.06)| 7.16(0.01)   | 7.16(0.00)   | 7.44(0.26)   | 7.17(0.03)    |

Table 7: Datasets used in experiments.

|                                      | # Feature $D$ | # Observed values $N$ | Tensor size $|\Omega_I|$ | Sparsity $N/|\Omega_I|$ |
|--------------------------------------|---------------|-----------------------|--------------------------|-------------------------|
| Solarflare (mis, 1989)               | 9             | 1067                  | 41472                    | 0.0257                  |
| SPECT (Kurgan et al., 2001)          | 23            | 267                   | 4194304                  | 1.7e-04                 |
| Lympho. (Zwitter & Soklic, 1988a)    | 18            | 148                   | 113246208                | 1.3e-06                 |
| Votes (mis, 1987)                    | 17            | 376                   | 86093442                 | 4.4e-06                 |
| Tumor (Zwitter & Soklic, 1988b)      | 17            | 301                   | 2654208                  | 1.1e-04                 |
| DMFT (Simonoff, 2003)                | 5             | 797                   | 2268                     | 0.352                   |
| Led7 (Olson et al., 2017)            | 8             | 3200                  | 1280                     | 2.500                   |
| Chess (Holte et al., 1989)           | 35            | 3196                  | >1.0e10                  | 3.2e-07                 |

$n = 1, \ldots, N$. Both the number of samples, $N$, and the sample dimension, $D$, vary across datasets as seen in Table 7. In the original datasets, each $i_d$ represents a categorical quantity such as color, location, gender, etc. By mapping these to natural numbers, each feature $i_d$ is converted to a natural number from 1 to $I_d$, where $I_d$ is the degree of freedom in the $d$-th feature. We randomly select 70% of the $N$ samples to create the training index set $\Omega^{\mathrm{train}}$, 15% of samples to create the validation index set $\Omega^{\mathrm{valid}}$, and the final 15% of the samples form the test index set $\Omega^{\mathrm{test}}$. Some datasets may contain exactly the same samples. To deal with such datasets, we suppose that these indices sets may contain multiple identical elements.

We create empirical tensors $\mathcal{T}^{\mathrm{train}}$, $\mathcal{T}^{\mathrm{valid}}$, and $\mathcal{T}^{\mathrm{test}}$, where each value $\mathcal{T}_i^\ell$ is defined as the number of $i$ in the set $\Omega^\ell$ for $\ell \in \{\mathrm{train}, \mathrm{valid}, \mathrm{test}\}$. They are typically very sparse tensors. The above procedure to create empirical tensors is consistent with the discussion at the beginning of Section 4. We normalize each tensor by dividing all the elements by the sum of the tensor to map them to a discrete probability distribution.

During the training phase, by optimizing the log-likelihood

$$D(\mathcal{T}^{\mathrm{train}}, \mathcal{P}) = \sum_{i \in \Omega^{\mathrm{train}}} \mathcal{T}_i^{\mathrm{train}} \log \mathcal{P}_i, \tag{30}$$

we decompose the tensor $\mathcal{T}^{\mathrm{train}}$ to obtain the reconstructed tensor $\mathcal{P}$, which approximates $\mathcal{T}^{\mathrm{train}}$. We adjust hyper-parameters such as tensor ranks, bounds, and learning rates to minimize the distance $D(\mathcal{T}^{\mathrm{valid}}, \mathcal{P})$. Finally, we evaluate the generalization error $D(\mathcal{T}^{\mathrm{test}}, \mathcal{P})$, where $\mathcal{P}$ is the reconstruction approximating $\mathcal{T}^{\mathrm{train}}$ with tuned rank. The proposed method and the baseline method have initial value dependence. Hence, all calculations were repeated five times to evaluate the mean and standard deviation of the negative log-likelihood per sample. In some baseline methods, CP, NNCP, NNCPHALS, Tucker, NNTucker, and NNTuckerHALS, the Frobenius norm is optimized instead of Equation (30). Thus, the nonnegativity or normalization is not ensured for the reconstruction in these baselines. We describe heuristics for these issues in Section D.2.2.

## D.2 IMPLEMENTATION DETAIL

We describe the implementation details of the proposed and baseline methods in the following. All experiments other than KLNTDMU and KLCPMU are conducted by Python 3.12.3. For KLNTDMU

and KLCPMU, we used Python 3.10.3 which is detailed in Section D.2.2. We provide our source code for all experiments in the supplementary material.

### D.2.1 PROPOSED METHOD

The pseudocodes for proposed methods are described in Algorithms 1 and 2. All tensors and mixture ratios were initialized with a uniform distribution between 0 and 1 and normalized as necessary, except for the experiments in Section C.6. The algorithm was terminated when the number of iterations of the EM step exceeded 1200 or when the difference of the log-likelihood from the previous iteration was below 10e-6. We manually determined the values of the ranks to be searched for each dataset so ensure that we observe underfitting, better fitting, and overfitting regimes for each validation dataset. In the EM tensor-train model, the ranks of the central core tensors are adjusted to be equal to or larger than the ranks of the core tensors at the edges. This is because modes with large mutual information are gathered in the center of the train-structure due to reordering. The searched rank ranges are available in `exp_config.py` in the supplementary material. To reorder tensor modes, we use the greedy method, which can be found in `MI.py` in the supplementary material.

### D.2.2 BASELINES

**BM, MPS, and LPS**    We downloaded the source code for Positive Matrix Product State (MPS), Real Born Machine (BM), and Real Locally Purified State (LPS) from the official repository. [2] The license of the code is described in the repository as MIT License. They optimize real-valued tensors and square each element to obtain nonnegative tensors. Each element of these real-valued tensors is initialized with the standard normal distribution. We varied the learning rates from 1.0e-4, 1.0e-3, ... to 1.0 for training data. We then used the learning rate that yielded the smallest validation score. According to the description in the `README` file, the batch size was fixed at 20, and the number of iterations was set to 10,000. We performed each experiment five times with each bond and evaluated the mean and standard deviation. We varied the bond of the model as 1, 2, ..., 8, and evaluated the test data with the bond that best fit the validation data.

**CNMFOPT**    We implemented CNMFOPT according to the original paper (Ibrahim & Fu, 2021). We iteratively optimize the factor matrices $A^{(1)}, \ldots, A^{(d)}$, and the weights $\lambda$ one after the other. We use the exponentiated gradient method (Bubeck, 2015) to optimize each factor matrix and weight. Although this optimization can be performed by closed-form update rules, all parameters cannot be updated simultaneously. Thus a loop is required for each update. This loop is called the inner iteration, which is repeated 100 times. The iterations of updating $(A^{(1)}, \ldots, A^{(d)}, \lambda)$ described above are called outer iteration. In the update rule for the exponential gradient method, the product of the derivative and the learning rate $\alpha$ is included in the exponential function. The learning rate $\alpha$ was selected from 0.005, 0.001, 0.0005, 0.0001, and 0.00005 to minimize the validation error. Learning with a larger learning rate was not feasible because it caused an overflow of the exponential function. The initial values of the parameters follow a uniform random distribution. The algorithm terminates when one of the following conditions is met: (1) We compute the KL divergence from the input data to the reconstruction after updating all factor matrices and weights. The change is less than 1.0e-4 compared to the previous outer loop. (2) The number of outer iterations exceeds 600. When the condition (2) is met, the algorithm performs up to $60000(D + 1)$ inner iterations in total. For the Chess dataset with $D = 35$, the number of iterations is large. Thus we set the number of inner loops to 20, and we terminated the computation after 120 outer loops for the Chess dataset.

**CP, Tucker, and Train**    We used the `parafac`, `tucker`, and `tensor_train` functions in Tensorly 0.6. (Kossaifi et al., 2019) for the CP, Tucker, and Train decompositions, respectively. The ranks of these decompositions were tuned within the same range as the proposed EMCP, EMTucker, and EMTrain decomposition. The parafac function includes the computation of a pseudo-inverse matrix, which leads to instability for sparse input tensors. Therefore, we added random values sampled from a uniform distribution from 0 to 1.0e-6 to all the elements of the histogram and then normalized the tensor $\mathcal{T}_i^{\text{train}}$ to stabilize the decomposition. The reconstructed tensors by CP, Tucker, and Train decompositions can have negative values. If we replace negative values with 0, we suffer from the NaN error in computing the cost function that includes the logarithmic function. Therefore,

---

[2]`https://github.com/glivan/tensor_networks_for_probabilistic_modeling`,

we replaced the negative values with the small value, 1.0e-9. After addressing the negative values described above, the reconstruction tensors were normalized, and we evaluated the negative log-likelihood. For a fair comparison, we set the convergence threshold to 1.0e-6, which is the same as the proposed methods. The maximum number of iterations was set to 250, which is 2.5 times larger than the default value.

**NNCP, NNCPHALS, NNTucker, and NNTuckerHALS** We used functions non_negative_parafac, non_negative_parafac_hals, non_negative_tucker_hals and non_negative_tucker_hals in Tensorly 0.6 for NNCP, NNCPHALS, NNTucker, and NNTucker-HALS, respectively. The ranks of these decompositions were tuned within the same range as the proposed EMCP and EMTucker decomposition. The reconstructed tensors by these baseline methods satisfy nonnegativity but not normalization. Thus we follow the same procedure as for CP, Tucker, and Train for normalization, described above. The convergence threshold and maximum number of iterations also follow the description above.

**KLCPMU and KLNTDMU** We used the ntf_mu and ntd_mu functions in the Nonnegative Factorization Techniques Toolbox (Marmoret & Cohen, 2020) for KLCPMU and KLNTDMU, respectively. Since this library does not support Python 3.12, we used Python 3.10.4 to run them. We set beta=1 to optimize the KL divergence. The tolerance for the convergence was set to 1.0e-6, which is the same as the proposed methods. The maximum number of iterations was set to the default value of 1000. The ranks of the decomposition were tuned within the same range as the proposed EMCP and EMTucker, respectively. However, regarding the Votes dataset, we tuned the rank within the range of 1 to 10 because KLCPMU did not converge for higher ranks due to the expensive computational cost per iteration.

### D.3 ADDITIONAL INFORMATION FOR REPRODUCIBILITY

**Environment** Experiments were conducted on Ubuntu 20.04.1 with a single core of 2.1GHz Intel Xeon CPU Gold 5218 and 128GB of memory. This work does not require GPU computing. The total computation time for all experiments, including tuning the learning rate of the baselines, was less than 240 hours, using 88 threads of parallel computing.

**Dataset detail, license, and availability** We downloaded real-world datasets described in Table 7 through the Python package ucimlrepo and pmlb or GitHub repository [2]. UCI datasets, Solarflare, SPECT, Lymphography, Votes, and Tumor are licensed under a Creative Commons Attribution 4.0 International (CC BY 4.0) license, as seen on each web page in the UCI database. [1] The other two datasets, Led7 and DMFT, are licensed under MIT license as seen in the official GitHub repository [3]. The imported data were directly converted to tensors by the procedure described in Section D.1.

---

[3] https://github.com/EpistasisLab/pmlb

