# OpenReview forum: "Non-negative Tensor Mixture Learning for Discrete Density Estimation"
_ICLR.cc/2025/Conference — Submitted to ICLR 2025_

### Official Review · Reviewer_edxN · 2024-11-03

**Soundness:** 2
**Presentation:** 2
**Contribution:** 3
**Rating:** 5
**Confidence:** 3

**Summary:**

The work deals with a nonnegative tensor decomposition algorithm for discrete density estimation. The framework is based on expectation maximization algorithm from KL divergence minimization perspective. Unlike the existing KL divergence-based nonnegative tensor decomposition algorithms for this problem that used gradient-based iterative steps for parameter updates, the proposed approach leverages the insights from the many-body approximation problem. Consequently, the updates of the parameters in the M-step boils down to closed form expressions, reminiscent of the updates derived in (Huang & Sidiropoulos,2017; Yeredor & Haardt, 2019) .

**Strengths:**

Strengths:

1.	The proposed tensor decomposition method is a general framework that can handle different types of tensor decomposition models like CP, Tucker and tensor train, innovatively utilizing the many body approximation technique from the (Ghalamkari et al.,2023). The closed-form updates of the parameters in the M-step of the EM algorithm is also attractive as it avoid tuning of any hyperparameters, which is often practically inconvenient

**Weaknesses:**

Weakness:

1.	The paper organization has lot of room for improvement. From the start of the paper, a clear description of problem is missing. Section 4 introduces the problem statement, while section 3 starts describing the approach (many-body approximation). Then, it is confusing to connect and understand that some of the terms have been obtained from the previous step of the algorithm.

2.	The problem formulation and the application of the solution itself is debatable. The tensor ${\cal T}$ is an empirical distribution tensor which is of the dimension of the categorical features. The entire solution of discrete density learning depends on the accuracy of the empirical tensor, which is hard to make sure due to curse of dimensionality.

3.	The problem is hardly scalable as it deals with the many-body approximation tensor ${\cal M}$ whose dimension is much larger than that of even the empirical tensor ${\cal T}$.

4.	The experiments and major baselines are lacking. In discrete density estimation, there are works using second and third order marginals, that can mitigate the curse of dimensionality problem to an extent.

a.	Kargas, Nikos & Sidiropoulos, N.D. & Fu, Xiao. (2017). Tensors, Learning, and “Kolmogorov Extension” for Finite-Alphabet Random Vectors. IEEE Transactions on Signal Processing. 66. 10.1109/TSP.2018.2862383.

b.	S. Ibrahim and X. Fu, "Recovering Joint Probability of Discrete Random Variables From Pairwise Marginals," in IEEE Transactions on Signal Processing, vol. 69, pp. 4116-4131, 2021, doi: 10.1109/TSP.2021.3090960

Discussions and comparisons with these baselines would help readers understand the strength of the approach (if any).

**Questions:**

Questions/Comments:

1.	Writing and Organization: It is important to introduce the low-rank tensor structures when discussing it in the introduction.

2.	It is commented that “it always converges regardless of the choice of the low-rank structure assumed in the model.” EM convergence is hard to establish unless it is well initialized. Hence, it is not easy to claim convergence for the proposed method

3.	While there is a technique introduced to reduce the computational complexity of tensor train decomposition as presented in Section 4.2, computational complexity of Tucker remains the same which becomes dominant especially the case of K>1 mixture cases. The convergence speed curves should be presented with time in x axis to understand where it stands with respect to the baselines. Due to the computational complexity, it is also challenging to utilize the mixture model in practice. Once may need to choose the component of the mixtures (the type of tensor decomposition) , rather than allowing the model to learn by itself. A detailed discussion on the limitation of the approach (in the main section) would help here.

4.	It is unclear how does the adaptive noise term guarantees the ``convexity” and convergence as claimed in Section 4.4. What do you mean by convexity here? How do you learn the noise parameter here? Do you learn different noise parameter for different low-rank tensor models?

5.	Experiments are limited to showing negative log likelihood. While it shows the dynamics of the algorithm, it does not show how does the approach learn the ground-truth. Simulation studies would help understand how well the method learns the true discrete density. In real data experiments, missing value prediction should be a better approach to understand the applicability of the approach.

6.	Minor typos: “distribution underlying the data” in Page 5, notation $\eta^k$ is confusing with $\eta$ raised to kth power.

---

> ### Author Response · Authors · 2024-11-22
> **Response to Reviewer edxN - Part 1**
>
> Thank you for your review. We provide our point-by-point response to each of your comments in the following.
>
> **Weaknesses:**
>
> > 1. The paper organization has lot of room for improvement. From the start of the paper, a clear description of problem is missing. Section 4 introduces the problem statement, while section 3 starts describing the approach (many-body approximation). Then, it is confusing to connect and understand that some of the terms have been obtained from the previous step of the algorithm.
> >
>
> Thank you for the nice suggestion. We have added the problem statement at the end of the introduction (Lines 105-110).
>
> > 2. The problem formulation and the application of the solution itself is debatable. The tensor is an empirical distribution tensor which is of the dimension of the categorical features. The entire solution of discrete density learning depends on the accuracy of the empirical tensor, which is hard to make sure due to curse of dimensionality.
> >
>
> We conducted additional experiments with synthetic data, demonstrating that the proposed method can estimate the true distribution as the sample size increases. (Please refer to Figure 7.)
>
>
> > 3. The problem is hardly scalable as it deals with the many-body approximation tensor M whose dimension is much larger than that of even the empirical tensor.
> >
>
> It is not true since the tensor M is also sparse under the assumption that the given tensor T is sparse (Line 373), and we do not have to get a summation of all values in M as discussed in Section 4.2. For example, the complexity of the EMCPTrain is O(IDNR^2) for tensor order D, the rank R, the number of samples N, and the degree of freedom of category I. Although we agree EMTucker is not scalable, this is not a drawback specific to our optimizing framework but rather a general issue with Tucker structures since it has a dense core tensor G. Our mixture EMCPTrain successfully works on higher-order tensors, such as the 35th-dimensional Chess dataset.
>
>
> > 4. The experiments and major baselines are lacking. In discrete density estimation, there are works using second and third order marginals, that can mitigate the curse of dimensionality problem to an extent. Discussions and comparisons with these baselines would help readers understand the strength of the approach (if any).
> >
>
> We added the pairwise marginalized-based CNMFOPT, which is developed in the suggested paper [b], as a baseline (Please refer to Table 1). The advantages of our approach over the suggested methods are as follows:
>
> - [Flexible modeling] Our approach allows flexible modeling, including CP, Tucker, Train, mixture, and noise terms, while their approach is focused on only CP formats.
> - [KL divergence optimization] We optimize the KL divergence while paper [a]. optimizes the L2 norm, which is not a natural measure of probability.
> - [Monotonic decreasing error function] Our algorithm is guaranteed to be monotonically decreasing of the objective function.
> - [No hyperparameter for optimization] While ADMM requires a hyperparameter ρ to avoid computation of the inverse of a nonregular matrix and the mirror descent algorithm requires the step size $\alpha$, our framework does not require such a hyperparameter for optimization.
>
> We added the above discussion in Section 2 in the revised version of our manuscript (Lines 138 - 143).
>
>
> **Questions/Comments:**
>
> > 1. Writing and Organization: It is important to introduce the
> low-rank tensor structures when discussing it in the introduction.
> >
>
> Thank you for the suggestion. We added the following sentence in the first paragraph of the Introduction.
>
> `` There are numerous variations of tensor low-rank decompositions, such as CP (Hitchcock, 1927), Tucker (Tucker, 1966), and Tensor Train decompositions (Oseledets, 2011), which differ in the low-rank structure of the decomposed representation.''
>
> > 2. It is commented that “it always converges regardless of the choice of the low-rank structure assumed in the model.” EM convergence is hard to establish unless it is well initialized. Hence, it is not easy to claim convergence for the proposed method
> >
>
> We prove that our EM algorithm always converges. Please refer to Theorem 4 in the supplementary material.

---

> ### Author Response · Authors · 2024-11-22
> **Response to Reviewer edxN - Part 2**
>
> > 3. While there is a technique introduced to reduce the computational complexity of tensor train decomposition as presented in Section 4.2, computational complexity of Tucker remains the same which becomes dominant especially the case of K>1 mixture cases. Due to the computational complexity, it is also challenging to utilize the mixture model in practice.
> >
>
> Although we admit the Tucker decomposition has difficulty in computational complexity for large tensors, it is not necessary to include Tucker decomposition in the mixture. In practice, our mixture EMCPTrain successfully works on higher-order tensors, such as the 35th-order Chess dataset as its complexity does not include the R^D complexity induced by the Tucker core.
>
>
> > The convergence speed curves should be presented with time in x axis to understand where it stands with respect to the baselines.
> >
>
> We assume that this comment arises from concerns about the scalability of the proposed method, but again, our method is scalable with dimension D (except for the EMTucker). The motivation for the experiments in Figures 9 and 10 is not computational time but to examine the number of iterations required for convergence since it is often pointed out that the EM algorithm requires a lot of iterations to converge.
>
>
>
> > Once may need to choose the component of the mixtures (the type of tensor decomposition), rather than allowing the model to learn by itself. A detailed discussion on the limitation of the approach (in the main section) would help here.
> >
>
> We added the following sentence in Section 6 (Line 534).
>
> ``This flexibility raises the question of how to systematically determine each component of the mixture, presenting an intriguing direction for future research.``
>
> > 4. It is unclear how does the adaptive noise term guarantees the ``convexity” and convergence as claimed in Section 4.4. What do you mean by convexity here?
> >
>
> We really appreciate your careful reading of it. We meant by convexity that both M-steps and E-steps are convex optimization problems, respectively. However, since it appears as if the entire algorithm is a convex optimization, we removed the word.
>
> > How do you learn the noise parameter here?
> >
>
> The noise parameter is the weight of the constant tensor, and it is learned by Equation (12), in the same way as the weights of other low-rank tensors.
>
> > Do you learn different noise parameter for different low-rank tensor models?
> >
>
> No. We assume that each model has only one noise term.
>
> > 5. Experiments are limited to showing negative log likelihood. While it shows the dynamics of the algorithm, it does not show how does the approach learn the ground-truth. Simulation studies would help understand how well the method learns the true discrete density.
> >
>
> We conducted additional experiments with synthetic data, demonstrating that the proposed method can estimate the true distribution as the sample size increases. (Please refer to Figure 7.)
>
> > In real data experiments, missing value prediction should be a better approach to understand the applicability of the approach.
> >
>
> Thank you for your suggestion. Missing value predictions heavily depend on the missingness process invoked, and missing completely at random might be unrealistic. We, therefore, conducted additional experiments to estimate synthetic true low-rank distributions from obtained samples. (Please refer to Figure 7.)

---

> > ### Comment · Reviewer_edxN · 2024-11-29
> > **Response to Authors**
> >
> > I thank the authors for the point-to-point response. I appreciate the generalizability of the method. But I have still some concerns about a few aspects, especially experiments with real datasets. Even though, the method is shown to estimate the pdf in synthetic experiments, there are no real usecases (e.g., classification, completion etc) shown to establish the merit of the method. I am deciding keep my score.

---

> > > ### Author Response · Authors · 2024-12-03
> > > **Response to Reviewer edxN**
> > >
> > > Thank you for your response.
> > >
> > > > there are no real usecases (e.g., classification, completion etc) shown to establish the merit of the method.
> > > >
> > >
> > > Regarding this point, we provide additional results for classification. Please refer to [General Response Part II](https://openreview.net/forum?id=mbo4YnWCHd&noteId=YveOn4Mpts).

---

### Official Review · Reviewer_Kjtx · 2024-11-03

**Soundness:** 2
**Presentation:** 3
**Contribution:** 3
**Rating:** 3
**Confidence:** 3

**Summary:**

This paper aims to parsimoniously unify a variety of low-rank tensor decomposition structures and address discrete density estimation using a novel mixture of decompositions method. The method permits mixtures of CP, Tucker, and tensor-train decompositions to model non-negative data. The authors derive a computationally efficient expectation-maximization (EM) algorithm for performing inference in their model. The inference algorithm allows for learning of the mixture weights of the components, which are tensor decompositions. In addition to establishing a unifying framework for low-rank tensor decompositions, the authors demonstrate their method’s effectiveness in estimating discrete mass functions by comparing a specific instance of their method, EMCPTrain, to a large body of existing ones. Their method achieves marginally better negative-log-likelihood per sample than existing methods.

**Strengths:**

The contribution of closed-form maximization updates for the specific EM algorithm is a strength. The computational efficiency, in some instances of the method (such as EMCPTrain), are useful. The unifying framework across low-rank tensor decompositions is parsimonious, and leveraging the parsimony of mixture models is a neat idea. I particularly like how the authors show that one may learn the weights of their mixture model, removing the need to choose between low-rank structures in advance of training.

**Weaknesses:**

The approach, while unifying in theory and a neat idea, yields very incremental empirical results at best. The empirical gains are modest. When more baselines are taken into account, as in Table 6, the gains are further reduced. The comparisons and evaluations are not well-organized, making the aggregate contribution difficult to evaluate across the many tables in Section 5 and Appendix C. There are some section of the paper that came across as unclear that first time I read it. In particular, the paper refers to “many-body approximation” many times in the first two sections of the paper without a clear definition. A formal definition in Section 1 would help clarify how the paper aims to leverage the many-body approximation representation.

**Questions:**

In my experience, adding a noise (or constant tensor) term to a non-negative tensor decomposition can significantly improve model fit, and the empirical results shown in Table 2 mostly demonstrate this phenomena. How much of the empirical improvements in negative log-likelihood are from the adaptive noise term? I view the ability to seamlessly learn the noise term from the data as an advantage of this method, although I am concerned it is the only substantial advantage of this method in practice.
When would it be useful to include a Tucker component in practice? EMTucker is computationally expensive, scaling as R^D. While EMTuckerN beats existing methods in Table 5, it generally performs worse than EMCPTrainN.

---

> ### Author Response · Authors · 2024-11-22
> **Response to Reviewer Kjtx**
>
> Thank you for your review. We provide our point-by-point response to each of your comments in the following.
>
>
> ### Weaknesses:
>
> > There are some section of the paper that came across as unclear that first time I read it. In particular, the paper refers to “many-body approximation” many times in the first two sections of the paper without a clear definition. A formal definition in Section 1 would help clarify how the paper aims to leverage the many-body approximation representation.
> >
>
> Thank you for your constructive suggestion. We added the definition of many-body approximation at the end of the introduction. (Lines 111 - 120)
>
>
> > The approach, while unifying in theory and a neat idea, yields very incremental empirical results at best. The empirical gains are modest. When more baselines are taken into account, as in Table 6, the gains are further reduced.
> >
>
> We agree that the empirical gains are modest in terms of generalization error. However, as you summarized in Strengths, we would like to emphasize that rather than simply achieving SOTA in density estimation, our gain is
>
> - a unified optimization framework that is flexible enough to be applied freely to a variety of non-negative low-rank structures (CP, Tucker, Train, …), mixtures, and adaptive noise-term,
> - convergence guarantee with monotonically decreasing error function,
> - closed update rules for all parameters simultaneously and,
> - eliminating the need to tune the learning rate and batch size, which is often required in modern methods such as CNMFOPT, MPS, BM, and LPS.
>
> > The comparisons and evaluations are not well-organized, making the aggregate contribution difficult to evaluate across the many tables in Section 5 and Appendix C.
> >
>
> We have added the following paragraph in Section C (Lines 1240-1269), which makes clear what we demonstrate in each experiment.
>
> ``The last part of this section is organized as follows. In Section C.1, we perform experiments on synthetic data to verify that the proposed algorithm can estimate the true distribution. In Section C.2, we observe that the adaptive noise term stabilizes the learning and significantly improves the validation error. In Sections C.3, C.4, and C.5, we see the generalized performance comparing different optimization methods for the same low-rank model while we compared the performance between different models in Table 1 in the main text. In Section C.6, we verify the number of EM iterations required for convergence since it is often pointed out that the EM algorithm requires a lot of iterations to converge (Ng et al., 2012; Chege et al., 2022)."
>
> ### Questions:
>
> > In my experience, adding a noise (or constant tensor) term to a non-negative tensor decomposition can significantly improve model fit, and the empirical results shown in Table 2 mostly demonstrate this phenomena. How much of the empirical improvements in negative log-likelihood are from the adaptive noise term? I view the ability to seamlessly learn the noise term from the data as an advantage of this method, although I am concerned it is the only substantial advantage of this method in practice.
> >
>
> As we can see from Figure 8, by introducing an adaptive noise term, the learning becomes stable, and the validation error (on the vertical axis) becomes quite small, which is consistent with your experience.
>
> > When would it be useful to include a Tucker component in practice?
> >
>
> The core tensor G in the Tucker structure describes the relationships between the factor matrices $A^1,\dots,A^D$. Thus, it is often used to analyze the relationships among factors.
>
> > EMTucker is computationally expensive, scaling as R^D.
> >
>
> As you pointed out, the Tucker decomposition has the drawback that it does not scale for large D because it involves a dense core tensor G. This is not a drawback specific to our optimizing framework, but rather a general issue with Tucker structures.
>
> > While EMTuckerN beats existing methods in Table 5, it generally performs worse than EMCPTrainN.
> >
>
> Tucker decomposition is also known to be prone to overfitting compared to Tensor Train. Due to these properties (overfitting and scalability), Tucker decomposition is not necessarily the first choice for density estimation, but it is frequently used for unsupervised learning to study relationships among factors, and our optimization method is applicable to those other tasks.

---

> > ### Comment · Reviewer_Kjtx · 2024-12-03
> >
> > I thank the authors for their response. However, I still see too many problems with the manuscript to increase my score. In particular, the experiments do not convince me of the method's usefulness in practice.

---

> > > ### Author Response · Authors · 2024-12-03
> > > **Response to Reviewer Kjtx**
> > >
> > > Thank you for your response.
> > >
> > > > The experiments do not convince me of the method's usefulness in practice.
> > >
> > > The practical advantages of our proposed method include its unified framework, flexibility, guaranteed convergence, a monotonic decrease in the KL divergence, and the elimination of hyperparameter tuning for optimization.
> > >
> > > We have also conducted additional experiments on classification to demonstrate the usefulness of our method. Please refer to [General Response Part II](https://openreview.net/forum?id=mbo4YnWCHd&noteId=YveOn4Mpts) for details.

---

### Official Review · Reviewer_JciN · 2024-11-07

**Soundness:** 3
**Presentation:** 3
**Contribution:** 2
**Rating:** 6
**Confidence:** 3

**Summary:**

This work presents an expectation-maximization (EM) based unified framework for nonnegative tensor decomposition that optimizes the Kullback-Leibler divergence, and further establishes a general relationship between low-rank decomposition and many-body approximation. The proposed framework offers a unified methodology for a variety of low-rank structures, including CP,
Tucker, and Train decompositions, and their combinations. A series of experiments are carried out to illustrate the merits of the developed methodology.

**Strengths:**

1. A unified methodology has been developed to deal with a variety of low-rank structures, including CP, Tucker, Train decompositions, and  their combinations, forming mixtures of low-rank tensors.
2. A mixture of low-rank tensor modeling procedure is developed  to empirically demonstrates inferential robustness and improved generalization.
3. Both theoretical analysis and numerical comparisons are provided to show the merits of the proposed methodology.

**Weaknesses:**

1. The convergence analysis seems overly coarse and somewhat redundant, as it only demonstrates that negative cross-entropy increases with iterations—an expected result given the maximization objective. More importantly, the analysis should address how negative cross-entropy could converge to the true value.
2. Please note that the computational cost of the proposed algorithm increases significantly with the size of the tensor. Kindly provide the specific order of computational complexity and compare it with the complexity orders of related algorithms.
3. The compared baselines are not SOTA. Please consider some recent methods for numerical comparisons.
4. Please provide a detailed description of the convergence conditions in Algorithm 1.

**Questions:**

Is there a specific general mathematical expression for the many-body approximation?

---

> ### Author Response · Authors · 2024-11-22
> **Response to Reviewer JciN**
>
> Thank you for your review. We provide our point-by-point response to each of your comments in the following.
>
> **Weaknesses:**
>
> > 1. The convergence analysis seems overly coarse and somewhat redundant, as it only demonstrates that negative cross-entropy increases with iterations—an expected result given the maximization objective.
> >
>
> The motivation of the analysis in Figures 9 and 10 is to verify the number of EM iterations required for convergence since it is often pointed out that the EM algorithm requires a lot of iterations to converge. We updated the description in Line 520 as
>
> ``The EM algorithm often requires a large number of iterations to converge. Thus, we conducted additional experiments in Section C.6 and confirmed that the proposed method converges with fewer iterations than the batch gradient method and with a similar number of iterations as the MU methods.''
>
> > More importantly, the analysis should address how negative cross-entropy could converge to the true value.
> >
>
> We agree with this point. Thus, we conducted additional experiments with synthetic data, demonstrating that the proposed method can estimate the true distribution as the sample size increases. (Please refer to Figure 7.)
>
> > 2. The compared baselines are not SOTA. Please consider some recent methods for numerical comparisons.
> >
>
> We added the recently developed CNMFOPT (Ibrahim & Fu, 2021) as a baseline. Please refer to Table 1 in the revised manuscript.
>
> > 3. Please note that the computational cost of the proposed algorithm increases significantly with the size of the tensor.
> >
>
> This is not true except for EMTucker since the tensor M is sparse, and the closed-form update requires summation on only observed indices. For example, the complexity of the EMCPTrain is O(IDNR^2) for tensor order D, the rank R, the number of samples N, and the degree of freedom of category I. In practice, our mixture model successfully works on large tensors, such as the 35th-dimensional Chess dataset. The complexity of the algorithm is discussed in Section 4.2.
>
>
> > Kindly provide the specific order of computational complexity and compare it with the complexity orders of related algorithms.
> >
>
> Sections 4.2 and C.6 describe the computational cost of the proposed methods and compare them to the baseline MPS. If you can clarify which specific algorithms you have in mind, we will be delighted to address them.
>
> > 4. Please provide a detailed description of the convergence conditions in Algorithm 1.
> >
>
> We do not need any conditions. Theorem 4 proves our algorithm *always* converges regardless of the low-rank structure and initial values.
>
>
> **Questions:**
> > Is there a specific general mathematical expression for the many-body approximation?
> >
>
> We added the definition of many-body approximation in the Introduction for readability (Lines 111-120).

---

> > ### Comment · Reviewer_JciN · 2024-11-29
> >
> > Thanks for the authors' response. I shall change my score to 6.

---

### Official Review · Reviewer_YenJ · 2024-11-25

**Soundness:** 3
**Presentation:** 3
**Contribution:** 2
**Rating:** 3
**Confidence:** 5

**Summary:**

This paper studies negative tensor decomposition that optimizes the Kullback-Leibler divergence. To avoid iterations in each M-step and learning rate tuning, they establish a general relationship between low-rank decomposition and many-body approximation. The framework offers not only a unified methodology for a variety of low-rank structures, including CP, Tucker, and Train decompositions, but also their combinations, forming mixtures of low-rank tensors. The weights of each low-rank tensor in the mixture can be learned from the data, which eliminates the need to carefully choose a single low- rank structure in advance

**Strengths:**

The logic of the paper is reasonable to me that they want to optimize each block in an alternating minimization/maximization way, the experiments look good and rich, demonstrating the benefit of the proposed algorithm.

**Weaknesses:**

I don't see novel contribution in this paper, to me, the nonnegative tensor factorization is well studied in both distance and divergence. The author find the optimal solutions by making use of EM or alternating method including weights $\eta$'s, which is well known.

**Questions:**

Is there any theoretical guarantee that your updating algorithm can converge to local minimal or global minimal. Though the objective is nonconvex, but there exists a wide class of factorization problem that local minimal is also global.

---

> ### Author Response · Authors · 2024-11-28
> **Response to Reviewer YenJ**
>
> > I don't see novel contribution in this paper, to me, the nonnegative tensor factorization is well studied in both distance and divergence. The author find the optimal solutions by making use of EM or alternating method including weights η's, which is well known.
> >
>
> Our contributions are as follows:
>
> 1. *[Novel modeling]* Our framework includes not only traditional low-rank structures (CP, Tucker, Train, …) but also novel modeling approaches, such as low-rank mixtures and adaptive noise terms keeping convergence guarantee.
> 2. *[Scalability]* The proposed methods worked on the Chess dataset, which is a 35-dimensional tensor (except for EMTucker).
> 3. *[Closed-form update]* We simultaneously update all parameters in the M-step and eliminate iteration in each M-step, thus there is no need for learning rate tuning.
>
> We emphasize that applying a well-known method (EM algorithm) to a well-studied problem (KL-based tensor factorization) is not necessarily trivial. In fact, a naive formulation of EM-based non-negative tensor factorization (EM-NTF) requires, as explained in lines 72 - 76 and 338 - 342, either alternating optimization to bound the log-likelihood (E-step) and maximizing each factor individually (M-step) or using gradient-based methods in each M-step.
>
> We overcome this by leveraging the closed-form solution of many-body approximation to simultaneously update all parameters at once in the M-step (Lines 342 - 343) and further leverage the sparseness of the given tensor T (Lines 370 - 374) to make the approach scalable.
>
> > Is there any theoretical guarantee that your updating algorithm can converge to local minimal or global minimal. Though the objective is nonconvex, but there exists a wide class of factorization problem that local minimal is also global.
> >
>
> The EM algorithm always converges to a stationary point. Please refer to Theorem 2 in the paper by C. F. Jeff Wu [1].
>
> [1] Wu, CF Jeff. "On the convergence properties of the EM algorithm." *The Annals of statistics* (1983): 95-103.

---

### Author Response · Authors · 2024-11-22
**General Response by Authors**

We appreciate all reviewers for their efforts. We summarize our rebuttal here. The major comments can be divided into the following five categories: convergence of the algorithm, scalability, readability, convergence to the true distribution, and comparison with SOTA.


**1. Convergence of the algorithm** (JciN, edxN)

*The proposed algorithm always converges*, which is proved by Theorem 4 regardless of initial value and low-rank structures.


**2. Scalability** (JciN, edxN)

*Our framework is scalable* except for EMTucker since the tensor M is sparse, and the closed-form update requires summation on only observed indices. In practice, our mixture model successfully works on large tensors, such as the 35th-dimensional Chess dataset.

**3. Readability** (Kjtx, edxN)

We inserted the definition of many-body approximation and problem setup in Section 1 (Lines 111-120).

**4. Convergence to the true distribution** (JciN)

We conducted additional experiments with synthetic data, demonstrating that the proposed method can estimate the true distribution as the sample size increases. (Please refer to Figure 7.)

**5. The empirical gain is limited, and there is no comparison with SOTA** (JciN, edxN)

We added the pairwise marginalized-based CNMFOPT as a baseline (Table 1). We agree that the empirical gains are modest in terms of generalization error. However, we emphasize our motivation is not SOTA and our gains are
- a unified optimization framework that is flexible enough to be applied freely to a variety of non-negative low-rank structures (CP, Tucker, Train, and Tree), mixtures, and adaptive noise-term,
- convergence guarantee monotonically decreasing the KL divergence,
- closed update rules for all parameters simultaneously,
- eliminating the need to tune the learning rate, which is often required in modern methods.


If you have any additional questions, concerns, or discussions, please feel free to respond.

Best,

Authors of the submission #6148

---

### Author Response · Authors · 2024-12-03
**General Response by Authors, Part II**

Given comments from Reviewers edxN and Kjtx, we provide the experiment for the classification based on the estimated density with additional five datasets.


Following the description in Section D.1, we estimate the density $p$. The final feature $x_D$ in the density $p(x_1,\dots,x_D)$ corresponds to the class label of the discrete item $(x_1,\dots,x_{D-1})$. Thus, for a given test sample ${\bf y} = (y_1,\dots,y_{D-1})$, we see the estimated density $p(x_1=y_1, \dots, x_{D-1}=y_{D-1}, c)$ for each class label $c$. The sample ${\bf y}$ is then classified into the class $c$ that maximizes $p(y_1,…,y_{D-1},c)$. We repeated the calculations five times with random initialization to evaluate the mean and standard error of accuracy.



|               | CNMFOPT    | MPS        | BM         | LPS        | EMCPTrainN |
| :--- | :---: |  :---: |  :---: |  :---: |  :---: |
| CarEvaluation | 0.71(0.00) | 0.92(0.03) | 0.79(0.12) | 0.97(0.00) | 0.91(0.02) |
| Mofn          | 0.77(0.04) | 0.97(0.02) | 0.99(0.00) | 1.00(0.00) | 0.98(0.01) |
| XD6           | 0.74(0.02) | 1.00(0.00) | 0.99(0.00) | 1.00(0.00) | 1.00(0.00) |
| Parity5p5     | 0.42(0.00) | 0.99(0.00) | 0.41(0.02) | 0.88(0.27) | 1.00(0.00) |
| GermanGSS     | 0.83(0.00) | 0.83(0.00) | 0.83(0.06) | 0.89(0.01) | 0.90(0.03) |

We emphasize that our method does not require hyperparameter for optimization, while other baselines, CNMFOPT, BM, MPS, and LPS, require learning rate and/or batch size tuning. Please refer to Section D.2.2 for the hyperparameter selection.

The experimental result demonstrates that the proposed method achieves competitive classification performance compared to existing methods.

Again, our motivation is not to achieve SOTA but to develop a generic, flexible, and unified framework for tensor factorization that optimizes the KL divergence with guaranteed convergence. Please also refer to our answer [**5**. in General Response Part I](https://openreview.net/forum?id=mbo4YnWCHd&noteId=uKnAvn0Co9).


---


### Dataset detail:

We used relatively large datasets in the main text to verify the scalability of our methods. However, these datasets have a small number of samples relative to the vast size of the sample space, making them unsuitable for evaluation in classification tasks (Table 7). Therefore, we used the following dataset for this experiment.

|               | D (dimension of tensor) | # Classes | # Samples | |
| :--- | :---: |  :---: |  :---: |  :---: |
| CarEvaluation | 7                       | 5       | 1728      | UCI Datasets               |
| Mofn          | 11                      | 2       | 1342      | PMLB Datasets               |
| XD6           | 10                      | 10      | 973       | PMLB Datasets               |
| Parity5p5     | 6                       | 4       | 400       | PMLB Datasets               |
| GermanGSS     | 6                       | 2       | 400       | PMLB Datasets               |

---

### Meta-Review · Area_Chair_5hR7 · 2024-12-20

**Metareview:**

This paper introduces an EM-based unified framework for tensor decomposition founded on KL-divergence minimization. The authors use low-rank decomposition and many-body approximations to derive closed-form updates during the M-step. The reviewers found the overall idea interesting and appreciated both the framework's applicability to various tensor decomposition scenarios and the absence of required hyperparameter tuning. However, they expressed concerns regarding the experimental validation of the proposed approach. Although the authors attempted to address these concerns during the rebuttal phase, they did not provide sufficiently convincing arguments or demonstrate clear practical benefits. I believe the current version of the paper requires significant revision, particularly in terms of experimental validation. Therefore, I recommend rejecting the paper in its current form.

**Additional Comments On Reviewer Discussion:**

During the rebuttal phase, the authors made an effort to address the reviewers' concerns, offering clarifications on the convergence analysis and the behavior of the proposed algorithm (specifically addressing the points raised by reviewers JciN and edxN). They also responded to the experimental validation issues highlighted by reviewers JciN, edxN, and Kjtx, adding additional experimental results. However, these results still demonstrated only minor practical benefits, failing to convince the reviewers of the approach’s merits.

---

### Decision · Program_Chairs · 2025-01-22

Reject